# DreamPRM: Domain-Reweighted Process Reward Model for Multimodal Reasoning

**Qi Cao**
University of California, San Diego
`q9cao@ucsd.edu`

**Ruiyi Wang**
University of California, San Diego
`ruiyi@ucsd.edu`

**Ruiyi Zhang**
University of California, San Diego
`ruz048@ucsd.edu`

**Sai Ashish Somayajula**
University of California, San Diego
`ssomayaj@ucsd.edu`

**Pengtao Xie**
University of California, San Diego
Mohamed bin Zayed University of Artificial Intelligence (MBZUAI)
`p1xie@ucsd.edu`

Project Page: `https://github.com/coder-qicao/DreamPRM`

## Abstract

Reasoning has substantially improved the performance of large language models (LLMs) on complicated tasks. Central to the current reasoning studies, Process Reward Models (PRMs) offer a fine-grained evaluation of intermediate reasoning steps and guide the reasoning process. However, extending PRMs to multimodal large language models (MLLMs) introduces challenges. Since multimodal reasoning covers a wider range of tasks compared to text-only scenarios, the resulting distribution shift from the training to testing sets is more severe, leading to greater generalization difficulty. Training a reliable multimodal PRM, therefore, demands large and diverse datasets to ensure sufficient coverage. However, current multimodal reasoning datasets suffer from a marked quality imbalance, which degrades PRM performance and highlights the need for an effective data selection strategy. To address the issues, we introduce DreamPRM, a domain-reweighted training framework for multimodal PRMs which employs bi-level optimization. In the lower-level optimization, DreamPRM performs fine-tuning on multiple datasets with domain weights, allowing the PRM to prioritize high-quality reasoning signals and alleviating the impact of dataset quality imbalance. In the upper-level optimization, the PRM is evaluated on a separate meta-learning dataset; this feedback updates the domain weights through an aggregation loss function, thereby improving the generalization capability of trained PRM. Extensive experiments on multiple multimodal reasoning benchmarks covering both mathematical and general reasoning show that test-time scaling with DreamPRM consistently improves the performance of state-of-the-art MLLMs. Further comparisons reveal that DreamPRM's domain-reweighting strategy surpasses other data selection methods and yields higher accuracy gains than existing test-time scaling approaches. Notably, DreamPRM achieves a top-1 accuracy of 85.2% on the MATHVISTA leaderboard using the o4-mini model, demonstrating its strong generalization in complex multimodal reasoning tasks.

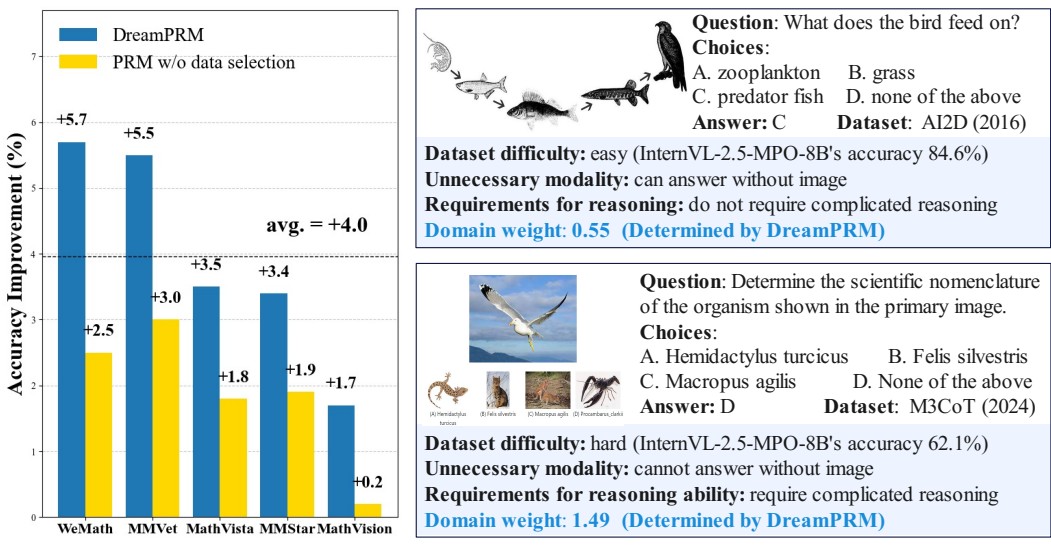

Figure 1: **DreamPRM improves multimodal reasoning by mitigating the dataset quality imbalance problem. Left**: On five benchmarks, DreamPRM outperforms base model (InternVL-2.5-8B-MPO [67]) by an average of $+4.0\%$. DreamPRM also consistently surpasses Vanilla PRM trained without data selection. **Right**: Easy AI2D [23] questions (weight 0.55) vs. hard M3CoT [6] questions (weight 1.49) shows how DreamPRM prioritizes data that demand deeper reasoning - samples requiring knowledge from both textual and visual modalities for step-by-step logical deduction.

# 1 Introduction

Reasoning [55] has significantly enhanced the logical and critical thinking capabilities of large language models (LLMs) [2, 8, 59, 49]. Post-training [45, 10] and test-time scaling strategies [44] enable sophisticated reasoning behaviors in LLMs and extend the length of Chain-of-Thoughts (CoTs) [71], thereby achieving strong results on challenging benchmarks [80, 47]. A key component of these advances is the Process Reward Models (PRMs) [29, 27], which provide fine-grained, step-wise supervision of the reasoning process and reliable selection of high-quality reasoning trajectories. These developments are proven highly effective for improving the performance of LLMs in complex tasks [38, 61].

Given the success with LLMs, a natural extension is to apply PRMs to multimodal large language models (MLLMs) [72, 28] to enhance their reasoning abilities. Early studies of multimodal PRMs demonstrate promise results, yet substantial challenges persist. Distinct from text-only inputs of LLMs, MLLMs must combine diverse visual and language signals: a high-dimensional, continuous image space coupled with discrete language tokens. This fusion dramatically broadens the input manifold and leads to more severe *distribution shifts* [56] from training to testing distributions. Consequently, directly utilizing PRM training strategies from the text domain [69, 37] underperforms, mainly due to the decreased generalizability [11] caused by the insufficient coverage of the multimodal input space.

A straightforward solution to this problem is to combine multiple datasets that emphasize different multimodal reasoning skills, thereby enlarging the sampling space. However, *quality imbalance* among existing multimodal reasoning datasets is more severe than in text-only settings: many contain noisy inputs such as unnecessary modalities [78] or questions of negligible difficulty [33], as illustrated in Fig. 1. Since these easy datasets contribute little to effective sampling, paying much attention to them can substantially degrade PRM performance. Therefore, an effective data selection strategy that filters out unreliable datasets and instances is crucial to training a high-quality multimodal PRM.

To overcome these challenges, we propose DreamPRM, a domain-reweighted training framework for multimodal PRMs. Inspired by domain-reweighting techniques [53, 12, 57], DreamPRM dynamically learns appropriate weights for each multimodal reasoning dataset, allowing them to contribute

unequally during training. Datasets that contain many noisy samples tend to receive lower domain weights, reducing their influence on PRM parameter updates. Conversely, high-quality datasets are assigned higher weights and thus play a more important role in optimization. This domain-reweighting strategy alleviates the issue of dataset quality imbalances. DreamPRM adopts a bi-level optimization (BLO) framework [14, 31] to jointly learn the domain weights and PRM parameters. At the lower level, the PRM parameters are optimized with Monte Carlo signals on multiple training domains under different domain weights. At the upper level, the optimized PRM is evaluated on a separate meta domain to compute a novel aggregation function loss, which is used to optimized the domain weights. Extensive experiments on a wide range of multimodal reasoning benchmarks verify the effectiveness of DreamPRM.

Our contributions are summarized as follows:

- We propose DreamPRM, a *domain-reweighted* multimodal process reward model training framework that dynamically adjusts the importance of different training domains. We formulate the training process of DreamPRM as a *bi-level optimization* (BLO) problem, where the lower level optimizes the PRM via domain-reweighted fine-tuning, and the upper level optimizes domain weights with an aggregation function loss. Our method helps address dataset quality imbalance issue in multimodal reasoning, and improves the generalization ability of PRM.

- We conduct extensive experiments using DreamPRM on a wide range of multimodal reasoning benchmarks. Results indicate that DreamPRM consistently surpasses PRM baselines with other data selection strategies, confirming the effectiveness of its bi-level optimization based domain-reweighting strategy. Notably, DreamPRM achieves a top-1 accuracy of 85.2% on the MATHVISTA leaderboard using the o4-mini model, demonstrating its strong generalization in complex multimodal reasoning tasks. Carefully designed evaluations further demonstrate that DreamPRM possesses both scaling capability and generalization ability to stronger models.

## 2  Related Works

**Multimodal reasoning**    Recent studies have demonstrated that incorporating Chain-of-Thought (CoT) reasoning [70, 25, 81] into LLMs encourages a step-by-step approach, thereby significantly enhancing question-answering performance. However, it has been reported that CoT prompting can't be easily extended to MLLMs, mainly due to hallucinated outputs during the reasoning process [67, 82, 19]. Therefore, some post-training methods have been proposed for enhancing reasoning capability of MLLMs. InternVL-MPO [67] proposes a mixed preference optimization that jointly optimizes preference ranking, response quality, and response generation loss to improve the reasoning abilities. Llava-CoT [74] creates a structured thinking fine-tuning dataset to make MLLM to perform systematic step-by-step reasoning. Some efforts have also been made for inference time scaling. RLAIF-V [77] proposes a novel self-feedback guidance for inference-time scaling and devises a simple length-normalization strategy tackling the bias towards shorter responses. AR-MCTS [11] combines Monte-Carlo Tree Search (MCTS) and Retrieval Augmented Generation (RAG) to guide MLLM search step by step and explore the answer space.

**Process reward model**    Process Reward Model (PRM) [29, 27, 38, 61] provides a more finer-grained verification than Outcome Reward Model (ORM) [9, 52], scoring each step of the reasoning trajectory. However, a central challenge in designing PRMs is obtaining process supervision signals, which require supervised labels for each reasoning step. Current approaches typically depend on costly, labor-intensive human annotation [29], highlighting the need for automated methods to improve scalability and efficiency. Math-Shepherd [64] proposes a method utilizing Monte-Carlo estimation to provide hard labels and soft labels for automatic process supervision. OmegaPRM [37] proposes a Monte Carlo Tree Search (MCTS) for finer-grained exploration for automatical labeling. MiPS [69] further explores the Monte Carlo estimation method and studies the aggregation of PRM signals.

**Domain-reweighting**    Domain reweighting methodologies are developed to modulate the influence of individual data domains, thereby enabling models to achieve robust generalization. Recently, domain reweighting has emerged as a key component in large language model pre-training, where

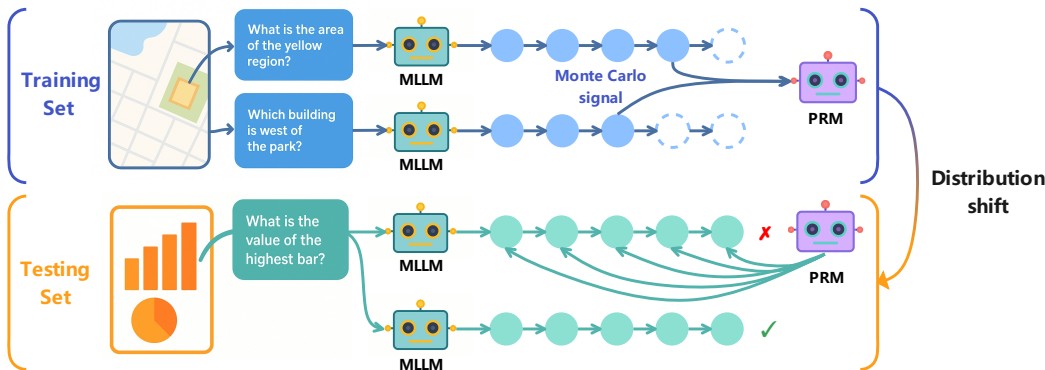

Figure 2: **General flow of training PRM and using PRM for inference. Training phase**: Train PRM with Monte Carlo signals from intermediate steps of Chain-of-Thoughts (CoTs). **Inference phase**: Use the trained PRM to verify CoTs step by step and select the best CoT. Conventional training of PRM has poor generalization capability due to *distribution shift* between training set and testing set.

corpora are drawn from heterogeneous sources. DoReMi [73] trains a lightweight proxy model with group distributionally robust optimization to assign domain weights that maximize excess loss relative to a reference model. DOGE [13] proposes a first-order bi-level optimization framework, using gradient alignment between source and target domains to update mixture weights online during training. Complementary to these optimization-based approaches, Data Mixing Laws [76] derives scaling laws that could predict performance under different domain mixtures, enabling low-cost searches for near-optimal weights without proxy models. In this paper, we extend these ideas to process supervision and introduce a novel bi-level domain-reweighting framework.

## 3   Problem Setting and Preliminaries

**Notations.**   Let $\mathcal{I}$, $\mathcal{T}$, and $\mathcal{Y}$ denote the multimodal input space (images), textual instruction space, and response space, respectively. A multimodal large language model (MLLM) is formalized as a parametric mapping $M_\theta : \mathcal{T} \times \mathcal{I} \to \Delta(\mathcal{Y})$, where $\hat{y} \sim M_\theta(\cdot|x)$ represents the stochastic generation of responses conditioned on input pair $x = (t, I)$ including visual input $I \in \mathcal{I}$ and textual instruction $t \in \mathcal{T}$, with $\Delta(\mathcal{Y})$ denoting the probability simplex over the response space. We use $y \in \mathcal{Y}$ to denote the ground truth label from a dataset.

The process reward model (PRM) constitutes a sequence classification function $\mathcal{V}_\phi : \mathcal{T} \times \mathcal{I} \times \mathcal{Y} \to [0, 1]$, parameterized by $\phi$, which quantifies the epistemic value of partial reasoning state $\hat{y}_i$ through scalar reward $p_i = \mathcal{V}_\phi(x, \hat{y}_i)$, modeling incremental utility toward solving instruction $t$ under visual grounding $I$. Specifically, $\hat{y}_i$ represents the first $i$ steps of a complete reasoning trajectory $\hat{y}$.

**PRM training with Monte Carlo signals.**   Due to the lack of ground truth epistemic value for each partial reasoning state $\hat{y}_i$, training of PRM requires automatic generation of approximated supervision signals. An effective approach to obtain these signals is to use the Monte Carlo method [69, 65]. We first feed the input question-image pair $x = (t, I)$ and the prefix solution $\hat{y}_i$ into the MLLM, and let it complete the remaining steps until reaching the final answer. We randomly sample multiple completions, compare their final answers to the gold answer $y$, and thereby obtain multiple correctness labels. PRM is trained as a sequence classification task to predict these correctness labels. The ratio of correct completions at the $i$-th step estimates the "correctness level" up to step $i$, which is used as the approximated supervision signals $p_i$ to train the PRM. Formally,

$$p_i = \texttt{MonteCarlo}(x, \hat{y}_i, y) = \frac{\texttt{num(correct completions from } \hat{y}_i)}{\texttt{num(total completions from } \hat{y}_i)} \tag{1}$$

**PRM-based inference with aggregation function.**   After training a PRM, a typical way of conducting PRM-based MLLM inference is to use aggregation function [69]. Specifically, for each candidate

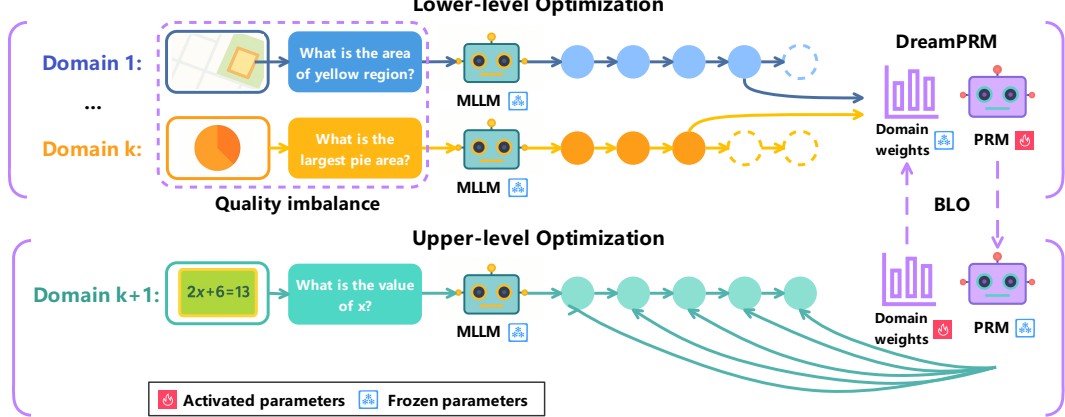

Figure 3: **The proposed bi-level optimization based domain-reweighting method. Lower-level optimization:** In this stage, PRM's parameters are updated on multiple datasets with domain weights, allowing the PRM to prioritize domains with better quality. **Upper-level optimization:** In this stage, the PRM is evaluated on a separate meta dataset to compute an aggregation function loss and optimize the domain weights. DreamPRM helps address dataset quality imbalance problems and leads to stronger and more generalizable reasoning performance.

solution $\hat{y}$ from the MLLM, PRM will generate a list of predicted probabilities $p = \{p_1, p_2, ..., p_n\}$ accordingly, one for each step $\hat{y}_i$ in the solution. The list of predicted probabilities are then aggregated using the following function:

$$\mathcal{A}(p) = \sum_{i=1}^{n} \log \frac{p_i}{1 - p_i}. \tag{2}$$

The aggregated value corresponds to the score of a specific prediction $\hat{y}$, and the final PRM-based solution is the one with the highest aggregated score.

**Bi-level optimization.** Bi-level optimization (BLO) has been widely used in meta-learning [14], neural architecture search [31], and data reweighting [54]. A BLO problem is usually formulated as:

$$\min_{\alpha} \mathcal{U}(\alpha, \phi^*(\alpha)) \tag{3}$$

$$s.t. \phi^*(\alpha) = \arg\min_{\phi} \mathcal{L}(\phi, \alpha) \tag{4}$$

where $\mathcal{U}$ is the upper-level optimization problem (OP) with parameter $\alpha$, and $\mathcal{L}$ is the lower-level OP with parameter $\phi$. The lower-level OP is nested within the upper-level one, and the two OPs are mutually dependent.

## 4 The Proposed Domain-reweighting Method

**Overview.** Training process reward models (PRMs) for MLLMs is challenging for two reasons: (1) dataset (domain) quality imbalance, and (2) discrepancy between training and inference procedures. To address these two challenges, we propose DreamPRM, which automatically searches for domain importance using a novel aggregation function loss that better simulates the inference process of PRM. Under a bi-level optimization framework, it optimizes PRM parameters with Monte Carlo signals at the lower level, and optimizes trainable domain importance weights with aggregation function loss at the upper level. An overview of DreamPRM method is shown in Fig. 3.

**Datasets.** We begin with $K+1$ datasets, each from a distinct domain (e.g., science, geometry). The first $K$ datasets form the training pool $\mathcal{D}_{\text{tr}} = \{\mathcal{D}_1, \ldots, \mathcal{D}_K\}$, while the remaining dataset, $\mathcal{D}_{\text{meta}} = \mathcal{D}_{K+1}$, is a meta (validation) dataset with better quality.

**Lower-level optimization: domain-reweighted training of PRM.** In lower-level optimization, we aim to update the weights $\phi$ of PRM with domain-reweighted training. We first define the typical PRM training loss $\mathcal{L}_{tr}$ on a single domain $\mathcal{D}_k$, given PRM parameters $\phi$, as follows:

$$\mathcal{L}_{tr}(\mathcal{D}_k, \phi) = \sum_{(x,y) \in \mathcal{D}_k} \sum_{i=1}^{n} \mathcal{L}_{MSE}(\mathcal{V}_\phi(x, \hat{y}_i), p_i) \tag{5}$$

where $\hat{y}_i$ is the prefix of MLLM generated text $\hat{y} = M_\theta(x)$ given input pair $x = (t, I)$, and $p_i$ is the process supervision signal value obtained by Monte Carlo estimation given input pair $x$, prefix $\hat{y}_i$ and ground truth label $y$, as previously defined in Equation 1. The PRM is optimized by minimizing the mean squared error (MSE) between supervision signal and PRM predicted score $\mathcal{V}_\phi(x, \hat{y}_i)$. With the PRM training loss on a single domain $\mathcal{D}_k$ above, we next define the domain-reweighted training objective of PRM on multiple training domains $\mathcal{D} = \{\mathcal{D}_k\}_{k=1}^{K}$. The overall objective is a weighted sum of the single-domain PRM training losses, allowing the contribution of each domain to be adjusted during the learning process:

$$\mathcal{L}_{tr}(\mathcal{D}_{tr}, \phi, \alpha) = \sum_{k=1}^{K} \alpha_k \mathcal{L}_{tr}(\mathcal{D}_k, \phi) \tag{6}$$

Here, $\alpha = \{\alpha_k\}_{k=1}^{K}$ represents the trainable domain weight parameters, indicating the importance of each domain. By optimizing this objective, we obtain the optimal value of PRM parameters $\phi^*$:

$$\phi^*(\alpha) = \arg\min_{\phi} \mathcal{L}_{tr}(\mathcal{D}_{tr}, \phi, \alpha) \tag{7}$$

It is worth mentioning that only $\phi$ is optimized at this level, while $\alpha$ remains fixed.

**Upper-level optimization: learning domain reweighting parameters.** In upper-level optimization, we optimize the domain reweighting parameter $\alpha$ on meta dataset $\mathcal{D}_{meta}$ given optimal PRM weights $\phi^*(\alpha)$ obtained from the lower level. To make the meta learning target more closely reflect the actual PRM-based inference process, we propose a novel meta loss function $\mathcal{L}_{meta}$, different from the training loss $\mathcal{L}_{tr}$. Specifically, we first obtain an aggregated score $\mathcal{A}(p)$ for each generated solution $\hat{y}$ from the MLLM given input pair $x = (t, I)$, following process in Section 3. We then create a ground truth signal $r(\hat{y}, y)$ by assigning it a value of 1 if the generated $\hat{y}$ contains ground truth $y$, and 0 otherwise. The meta loss is defined as the mean squared error between aggregated score and ground truth signal:

$$\mathcal{L}_{meta}(\mathcal{D}_{meta}, \phi^*(\alpha)) = \sum_{(x,y) \in \mathcal{D}_{meta}} \mathcal{L}_{MSE}(\sigma(\mathcal{A}(\mathcal{V}_{\phi^*(\alpha)}(x, \hat{y}))), r(\hat{y}, y)) \tag{8}$$

where $\mathcal{A}$ represents the aggregation function as previously defined in Equation 2, and $\sigma$ denotes the sigmoid function to map the aggregated score to a probability. Accordingly, the optimization problem at the upper level is formulated as follows:

$$\min_{\alpha} \mathcal{L}_{meta}(\mathcal{D}_{meta}, \phi^*(\alpha)) \tag{9}$$

To solve this optimization problem, we propose an efficient gradient-based algorithm, which is detailed in Appendix A.

# 5 Experimental Results

## 5.1 Experimental settings

**Multistage reasoning.** To elicit consistent steady reasoning responses from current MLLMs, we draw on the Llava-CoT approach [75], which fosters structured thinking prior to answer generation.

Table 1: **Comparative evaluation of DreamPRM and baselines on multimodal reasoning benchmarks. Bold numbers** indicate the best performance, while underlined numbers indicate the second best. The table reports accuracy (%) on five datasets: WEMATH, MATHVISTA, MATHVISION, MMVET, and MMSTAR.

| | Math Reasoning | | | General Reasoning | |
|---|---|---|---|---|---|
| | WEMATH *(loose)* | MATHVISTA *(testmini)* | MATHVISION *(test)* | MMVET *(v1)* | MMSTAR *(test)* |
| *Zero-shot Methods* | | | | | |
| Gemini-1.5-Pro [50] | 46.0 | 63.9 | 19.2 | 64.0 | 59.1 |
| GPT-4v [46] | 51.4 | 49.9 | 21.7 | **67.7** | 62.0 |
| LLaVA-OneVision-7B [26] | 44.8 | 63.2 | 18.4 | 57.5 | 61.7 |
| Qwen2-VL-7B [66] | 42.9 | 58.2 | 16.3 | 62.0 | 60.7 |
| InternVL-2.5-8B-MPO [67] | 51.7 | 65.4 | 20.4 | 55.9 | 58.9 |
| *Test-time Scaling Methods (InternVL-2.5-8B-MPO based)* | | | | | |
| Self-consistency [68] | 56.4 | 67.1 | 20.7 | 57.4 | 59.6 |
| Self-correction [17] | 54.0 | 63.8 | 21.6 | 54.9 | 59.7 |
| ORM [52] | 56.9 | 65.3 | 20.5 | 55.9 | 60.1 |
| Vanilla PRM [29] | 54.2 | 67.2 | 20.6 | 58.9 | 60.8 |
| CaR-PRM [16] | 54.7 | 67.5 | 21.0 | 60.6 | 61.1 |
| s1-PRM [44] | 57.1 | 65.8 | 20.2 | 60.1 | 60.4 |
| **DreamPRM (ours)** | **57.4** | **68.9** | **22.1** | 61.4 | **62.3** |

Specifically, we prompt MLLMs to follow five reasoning steps: `(1) Restate the question. (2) Gather evidence from the image. (3) Identify any background knowledge needed. (4) Reason with the current evidence. (5) Summarize and conclude with all the information.` We also explore zero-shot prompting settings in conjunction with structural reasoning, which can be found in Appendix C. We use 8 different chain-of-thought reasoning trajectories for all test-time scaling methods, unless otherwise stated.

**Base models.** For inference, we use InternVL-2.5-8B-MPO [67] as the base MLLM, which has undergone post-training to enhance its reasoning abilities and is well-suited for our experiment. For fine-tuning PRM, we adopt Qwen2-VL-2B-Instruct [66]. Qwen2-VL is a state-of-the-art multimodal model pretrained for general vision-language understanding tasks. This pretrained model serves as the initialization for our fine-tuning process.

**Training hyperparameters.** In the lower-level optimization, we perform 5 inner gradient steps per outer update (unroll steps = 5) using the AdamW [32] optimizer with learning rate set to $5 \times 10^{-7}$. In the upper-level optimization, we use the AdamW optimizer (lr = 0.01, weight decay = $10^{-3}$) and a StepLR scheduler (step size = 5000, $\gamma = 0.5$). In total, DreamPRM is fine-tuned for 10000 iterations. Our method is implemented with Betty [7], and the fine-tuning process takes approximately 10 hours on one NVIDIA A100 GPUs.

**Baselines.** We use three major categories of baselines: **(1)** State-of-the-art models on public leaderboards, including Gemini-1.5-Pro [50], GPT-4V [46], LLaVA-OneVision-7B [26], Qwen2-VL-7B [66]. We also carefully reproduce the results of InternVL-2.5-8B-MPO with structural thinking. **(2)** Test-time scaling methods (excluding PRM) based on the InternVL-2.5-8B-MPO model, including: (i) Self-consistency [68], which selects the most consistent reasoning chain via majority voting over multiple responses; (ii) Self-correction [17], which prompts the model to critically reflect on and revise its initial answers; and (iii) Outcome Reward Model (ORM) [52], which evaluates and scores the final response to select the most promising one. **(3)** PRM-based methods, including: (i) Vanilla PRM trained without any data selection, as commonly used in LLM settings [29]; (ii) s1-PRM, which selects high-quality reasoning responses based on three criteria - difficulty, quality,

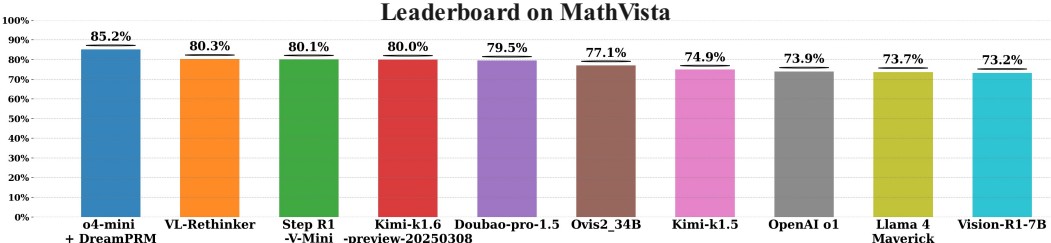

Figure 4: **Leaderboard on MathVista (as of October 15, 2025).** The first column ("o4-mini + DreamPRM") reports our own evaluation, while the remaining results are taken from the official MathVista leaderboard. The compared models include VL-Rethinker [62], Step R1-V-Mini [58], Kimi-k1.6-preview [43], Kimi-k1.5 [24], Doubao-pro-1.5 [60], Ovis2-34B [1], OpenAI o1 [45], Llama 4 Maverick [41, 42], and Vision-R1-7B [18].

and diversity - following the s1 strategy [44]; and (iii) CaR-PRM, which filters high-quality visual questions using clustering and ranking techniques, as proposed in CaR [16].

**Datasets and benchmarks.** We use 15 multimodal datasets for lower-level optimization ($\mathcal{D}_{tr}$), covering four domains: science, chart, geometry, and commonsense, as listed in Appendix Table 2. For upper-level optimization ($\mathcal{D}_{meta}$), we adopt the MMMU [79] dataset. Evaluation is conducted on five multimodal reasoning benchmarks: WeMath [48], MathVista [33], MathVision [63], MMVet [78], and MMStar [5]. Details are provided in Appendix B.

## 5.2 Benchmark evaluation of DreamPRM

Tab. 1 presents the primary experimental results. We observe that: **(1) DreamPRM outperforms other PRM-based methods**, highlighting the effectiveness of our domain reweighting strategy. Compared to the vanilla PRM trained without any data selection, DreamPRM achieves a consistent performance gain of 2%-3% across all five datasets, suggesting that effective data selection is crucial for training high-quality multimodal PRMs. Moreover, DreamPRM also outperforms s1-PRM and CaR-PRM, which rely on manually designed heuristic rules for data selection. These results indicate that selecting suitable reasoning datasets for PRM training is a complex task, and handcrafted rules are often suboptimal. In contrast, our automatic domain-reweighting approach enables the model to adaptively optimize its learning process, illustrating how data-driven optimization offers a scalable solution to dataset selection challenges. **(2) DreamPRM outperforms SOTA MLLMs with much fewer parameters**, highlighting the effectiveness of DreamPRM. For example, DreamPRM significantly surpasses two trillion-scale closed-source LLMs (GPT-4v and Gemini-1.5-Pro) on 4 out of 5 datasets. In addition, it consistently improves the performance of the base model, InternVL-2.5-8B-MPO, achieving an average gain of 4% on the five datasets. These results confirm that DreamPRM effectively yields a high-quality PRM, which is capable of enhancing multimodal reasoning across a wide range of benchmarks. **(3) DreamPRM outperforms other test-time scaling methods**, primarily because it enables the training of a high-quality PRM that conducts fine-grained, step-level evaluation. While most test-time scaling methods yield moderate improvements, DreamPRM leads to the most substantial gains, suggesting that the quality of the reward model is critical for effective test-time scaling. We further provide case studies in Appendix D, which intuitively illustrate how DreamPRM assigns higher scores to coherent and high-quality reasoning trajectories.

## 5.3 Leaderboard performance of DreamPRM

As shown in Fig. 4, DreamPRM achieves the **top-1 accuracy of 85.2%** on the MathVista leaderboard (as of October 15, 2025). The result (o4-mini + DreamPRM) has been officially verified through the MathVista evaluation. Compared with a series of strong multimodal reasoning baselines, including VL-Rethinker [62], Step R1-V-Mini [58], Kimi-k1.6-preview [43], Doubao-pro-1.5 [60], Ovis2-34B [1], OpenAI o1 [45], Llama 4 Maverick [41, 42], and Vision-R1-7B [18], DreamPRM demonstrates clearly superior multimodal reasoning capability.

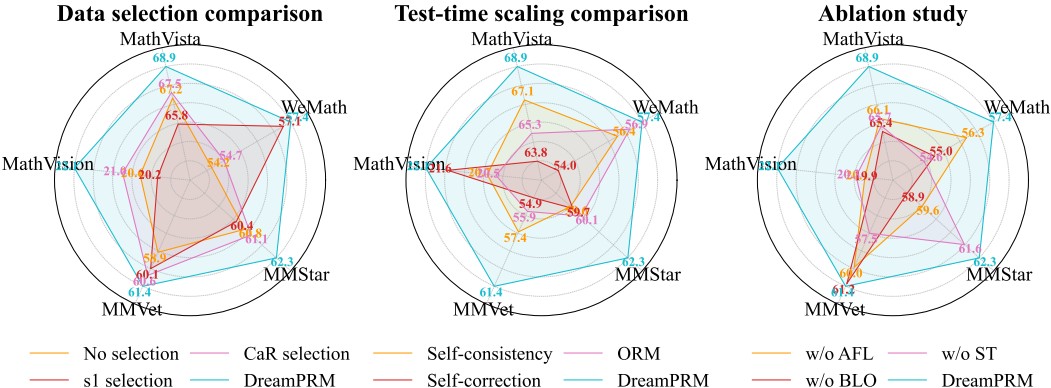

Figure 5: **Comparative evaluation of DreamPRM on multimodal reasoning benchmarks.** Radar charts report accuracy (%) on five datasets (WEMATH, MATHVISTA, MATHVISION, MMVET, and MMSTAR). **(a)** Impact of different data selection strategies. **(b)** Comparison with existing test-time scaling methods. **(c)** Ablation study of three key components, i.e. w/o aggregation function loss (AFL), w/o bi-level optimization (BLO), and w/o structural thinking (ST).

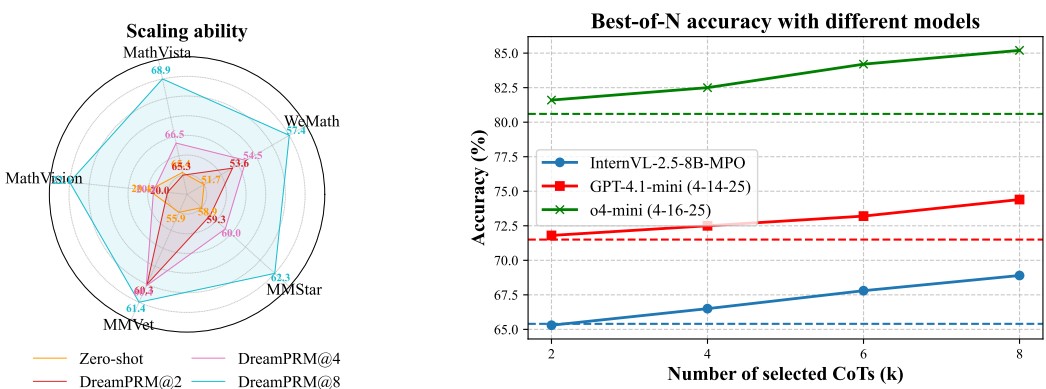

Figure 6: **Scaling ability and cross-model generalization.** **(a)** Radar chart of five multimodal reasoning benchmarks shows that DreamPRM delivers monotonic accuracy gains as the number of selected chains-of-thought increases (@2, @4, @8) over the pass@1 baseline. **(b)** Best-of-$N$ accuracy curves for InternVL-2.5-8B-MPO (blue), GPT-4.1-mini (red) and o4-mini (green) on MATHVISTA confirm that the same DreamPRM-ranked CoTs generalize across models, consistently outperforming pass@1 performance (dashed lines) as $k$ grows.

Table 5 in Appendix provides a detailed comparison among various Process Reward Model (PRM) variants built on the same o4-mini backbone. DreamPRM surpasses all counterparts, improving the base o4-mini model from 80.6% (pass@1) and 81.7% (self-consistency@8) to 85.2%. This consistent gain verifies the effectiveness of DreamPRM in enhancing reasoning accuracy through process-level supervision and reliable consensus across multiple chains of thought.

## 5.4 Scaling and generalization analysis of DreamPRM

**DreamPRM scales reliably with more CoT candidates.** As shown in the left panel of Fig. 6, the accuracy of DreamPRM consistently improves on all five benchmarks as the number of CoTs increases from $k=2$ to $k=8$, expanding the radar plot outward. Intuitively, a larger set of candidates increases the likelihood of including high-quality reasoning trajectories, but it also makes identifying the best ones more challenging. The consistent performance gains indicate that DreamPRM effectively verifies and ranks CoTs, demonstrating its robustness in selecting high-quality reasoning trajectories under more complex candidate pools.

**DreamPRM transfers seamlessly to stronger base MLLMs.** The right panel of Fig.6 shows the MATHVISTA accuracy when applying DreamPRM to recent MLLMs, GPT-4.1-mini *(2025-04-14)* [46] and o4-mini *(2025-04-16)* [45]. For o4-mini model, the pass@1 score of 80.6% steadily increases to 85.2% at $k=8$, surpassing the previous state-of-the-art performance. This best-of-$N$ trend, previously observed with InternVL, also holds for GPT-4.1-mini and o4-mini, demonstrating the generalization ability of DreamPRM. Full results of these experiments are provided in Tab. 3.

## 5.5 Ablation study

In this section, we investigate the importance of three components in DreamPRM: **(1)** bi-level optimization, **(2)** aggregation function loss in upper-level, and **(3)** structural thinking prompt (detailed in Section 5.1). As shown in the rightmost panel of Fig. 5, the complete DreamPRM achieves the best results compared to three ablation baselines across all five benchmarks. Eliminating bi-level optimization causes large performance drop (e.g., -3.5% on MATHVISTA and -3.4% on MMSTAR). Removing aggregation function loss leads to a consistent 1%-2% decline (e.g., 57.4% → 56.3% on WEMATH). Excluding structural thinking also degrades performance (e.g., -1.8% on MATHVISION). These results indicate that all three components are critical for DreamPRM to achieve the best performance. More detailed results are shown in Appendix Tab. 4.

## 5.6 Analysis of learned domain weights

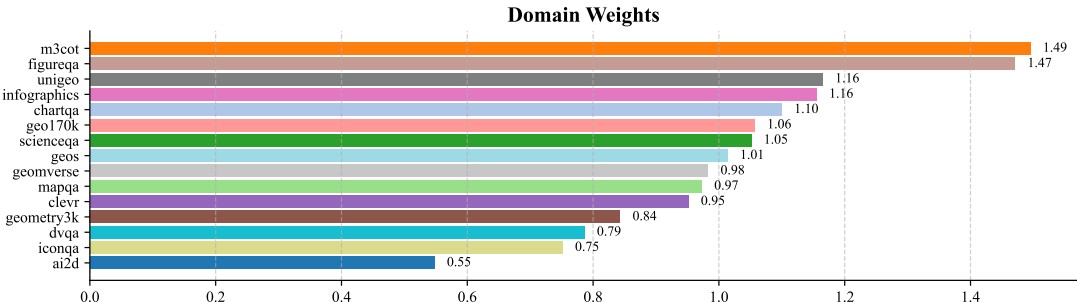

Figure 7: Learned domain weights after the convergence of the DreamPRM training process.

The final domain weights (Fig. 7) range from 0.55 to 1.49: M3COT [6] and FIGUREQA [21] receive the highest weights (approximately 1.5), while AI2D [23] and ICONQA [36] are assigned lower weights (less than 0.8). This learned weighting pattern contributes to improved PRM performance, indicating that the quality imbalance problem across reasoning datasets is real and consequential. Additionally, as shown in Fig. 9 in Appendix, all domain weights are initialized to 1.0 and eventually converge during the training process of DreamPRM.

## 6 Conclusions

We propose DreamPRM, the first domain-reweighted PRM framework for multimodal reasoning. By automatically searching for domain weights using a bi-level optimization framework, DreamPRM effectively mitigates issues caused by dataset quality imbalance and significantly enhances the generalizability of multimodal PRMs. Extensive experiments on five diverse benchmarks confirm that DreamPRM outperforms both vanilla PRMs without domain reweighting and PRMs using heuristic data selection methods. We also observe that the domain weights learned by DreamPRM correlate with dataset quality, effectively separating challenging, informative sources from overly simplistic or noisy ones. These results highlight the effectiveness of our proposed automatic domain reweighting strategy.

## Acknowledgments

This work was supported by the National Science Foundation (IIS2405974 and IIS2339216) and the National Institutes of Health (R35GM157217).

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

# Appendix

## A  Optimization algorithm

Directly solving the bi-level optimization problem in Equation 9 can be computational prohibitive due to its nested structure. Following previous work [7], we use approximated algorithm with a few unrolling steps. For example, under one-step unrolling, the updating of PRM's weights can be expressed as:

$$\phi^{(t+1)} = \phi^{(t)} - \beta_1 \nabla_\phi \mathcal{L}_{tr}(\mathcal{D}_{tr}, \phi, \alpha) \tag{10}$$

where $\beta_1$ is the learning rate in lower level optimization. After obtaining the updated PRM parameter $\phi^{(t+1)}$ from Equation 10, the domain-reweighting parameter $\alpha$ is then updated as follows:

$$\alpha^{(t+1)} = \alpha^{(t)} - \beta_2 \nabla_\alpha \mathcal{L}_{meta}(\mathcal{D}_{meta}, \phi^*(\alpha)) \tag{11}$$

where $\beta_2$ is the learning rate for upper level optimization. The two optimization steps in Equation 10 and Equation 11 are conducted iteratively until convergence to get optimal PRM weights $\phi^*$ and optimal domain reweighting parameter $\alpha^*$.

## B  Datasets and benchmarks

Table 2: Multimodal datasets involved in the fine-tuning of DreamPRM, organized by task category.

| Task | Dataset |
|---|---|
| Science | AI2D [23], ScienceQA [35], M3CoT [6] |
| Chart | ChartQA [39], DVQA [20], MapQA [3], FigureQA [21] |
| Geometry | Geo170k [15], Geometry3K [34], UniGeo [4], GeomVerse [22], GeoS [51] |
| Commonsense | IconQA [36], InfographicsVQA [40], CLEVR-Math [30] |

For datasets used in lower-level optimization ($\mathcal{D}_{tr}$ in Section 4), our study utilizes a diverse set of datasets, spanning multiple domains to ensure a comprehensive coverage of multimodal reasoning tasks, as reported in Tab. 2. The selected 15 multimodal datasets covers 4 major categories including science, chart, geometry and commonsense, with a wide range of task types (QA, OCR, spatial understanding). Additionally, we observe that for some questions, given the current structural thinking prompts, MLLMs consistently produce either correct or incorrect answers. Continuing to sample such questions is a waste of computational resources. Inspired by the dynamic sampling strategy in DAPO [78], we propose a similar dynamic sampling technique for Monte Carlo estimation that focuses on prompts with varied outcomes to improve efficiency. After processing and sampling, the training datasets in lower-level $\mathcal{D}_{tr}$ have around 15k examples (1k per each of the 15 domains), while the meta dataset in the upper-level $\mathcal{D}_{meta}$ has around 1k validation examples from the MMMU [79] dataset.

For the dataset used in upper-level optimization ($\mathcal{D}_{meta}$ in Section 4), we select data from MMMU [79] to simulate a realistic and diverse reasoning scenario. MMMU focuses on advanced perception and reasoning with domain-specific knowledge. Its questions span 30 subjects and 183 subfields, comprising 30 highly heterogeneous image types, such as charts, diagrams, maps, tables, music sheets, and chemical structures.

At evaluation time, we use five multimodal reasoning benchmarks for testing the capability of DreamPRM. WEMATH [48], MATHVISTA [33], and MATHVISION [63] focus more on math-related reasoning tasks and logic and critical thinking, while MMVET [78] and MMSTAR [5] focus more on real-life tasks that require common knowledge and general reasoning abilities.

# C  Structural Thinking Prompt

The detailed structural thinking prompt applied in our experiments is reported in Fig. 8. We carefully design 5 reasoning steps to boost the reasoning capabilities of the MLLMs and enable process supervision.

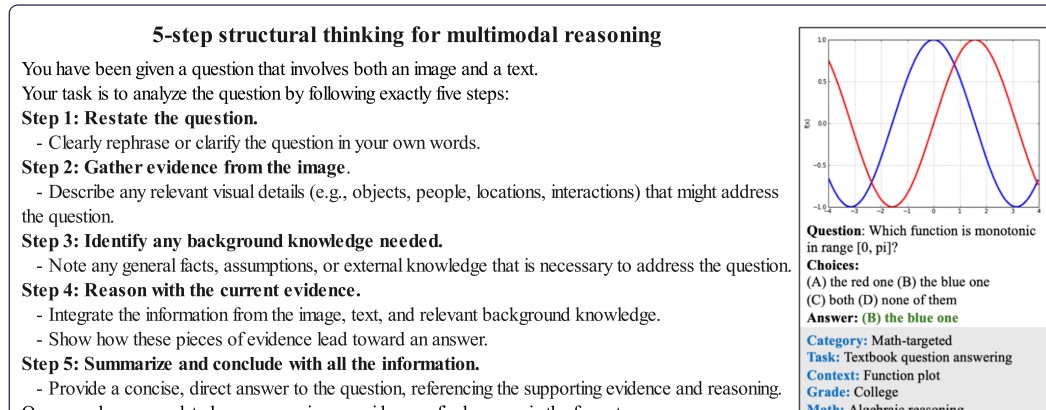

Figure 8: Zero-shot prompting for structural thinking.

Table 3: Accuracy on MathVista using DreamPRM with varying numbers $k$ of CoTs.

| Model Name | pass@1 | DreamPRM (select $k$ CoTs) | | | |
|---|---|---|---|---|---|
| | $k{=}1$ | $k{=}2$ | $k{=}4$ | $k{=}6$ | $k{=}8$ |
| InternVL-2.5-8B-MPO [67] | 65.4 | 65.3 | 66.5 | 67.8 | **68.9** |
| GPT-4.1-mini (4-14-25) [46] | 71.5 | 71.8 | 72.5 | 73.2 | **74.4** |

Table 4: Ablation study evaluating the impact of individual components of DreamPRM

| Ablation / Dataset | WeMath | MathVista | MathVision | MMVet | MMStar |
|---|---|---|---|---|---|
| **DreamPRM (original)** | **57.4** | **68.9** | **22.1** | **61.4** | **62.3** |
| w/o aggregation function loss | 56.3 (-1.1) | 66.1 (-2.8) | 20.1 (-2.0) | 60.0 (-1.4) | 59.6 (-2.7) |
| w/o bi-level optimization | 55.0 (-2.4) | 65.4 (-3.5) | 19.9 (-2.2) | 61.2 (-0.2) | 58.9 (-3.4) |
| w/o structural thinking | 54.6 (-2.8) | 65.7 (-3.2) | 20.3 (-1.8) | 57.5 (-3.9) | 61.6 (-0.7) |

# D  Additional Experimental Results

**Leaderboard performance details.** Table 5 presents a comprehensive comparison of different PRM variants built upon the same o4-mini backbone. DreamPRM consistently outperforms all baselines, elevating the base o4-mini performance from 80.6These steady improvements demonstrate the effectiveness of DreamPRM in enhancing reasoning accuracy through process-level supervision and promoting more reliable consensus across multiple chains of thought.

**Best-of-N results.** Tab. 3 reports the accuracy of two state-of-the-art models on MathVista dataset using DreamPRM with varying numbers $k$ of CoTs. The results indicate that the performance scales well with the number of CoTs.

**Ablation studies.** The exact results of ablation experiments in the main paper are included in Tab. 4, which emphasizes the importance of all the components in DreamPRM.

**Loss curves and domain weights.** The loss curves and domain weights during the fine-tuning of DreamPRM are illustrated in Fig. 9. It can be observed that the learnt distribution emphasizes

Table 5: Comparison of different PRM variants on the o4-mini model (evaluated on eight CoTs).

| Method | Accuracy |
|---|---|
| o4-mini | 80.6 |
| + Self-consistency | 81.7 |
| + ORM | 80.8 |
| + Vanilla-PRM | 84.2 |
| + DreamPRM | **85.2** |

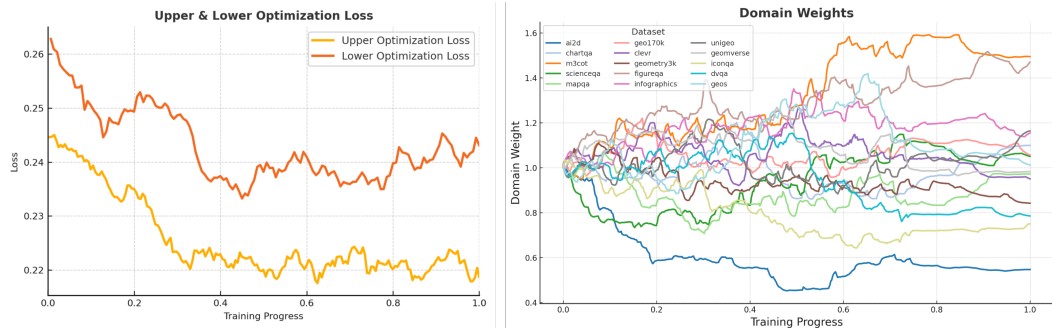

Figure 9: Optimization loss curves and dynamic domain weights throughout DreamPRM fine-tuning.

informative mathematical figure domains while attenuating less relevant sources. Additionally, domain weights start at 1.0 and quickly diverge, stabilizing after roughly half the training, and the inner and outer losses decrease steadily and plateau, indicating stable convergence of the bi-level training procedure.

**Case study.** A complete case study illustrating DreamPRM's step-wise evaluation is reported in Fig. 10. DreamPRM assigns higher scores to high-quality, coherent reasoning steps, while penalizes flawed or unsupported steps.

# E    Limitations & Future Work.

DreamPRM currently assumes a fixed set of domains and requires Monte-Carlo sampling, which can be computationally heavy. Future work could explore instance-level reweighting, adaptive sampling strategies, and integration with retrieval-augmented generation to further cut compute while broadening coverage. We will release code, trained weights, and evaluation scripts to facilitate reproducibility and community adoption.

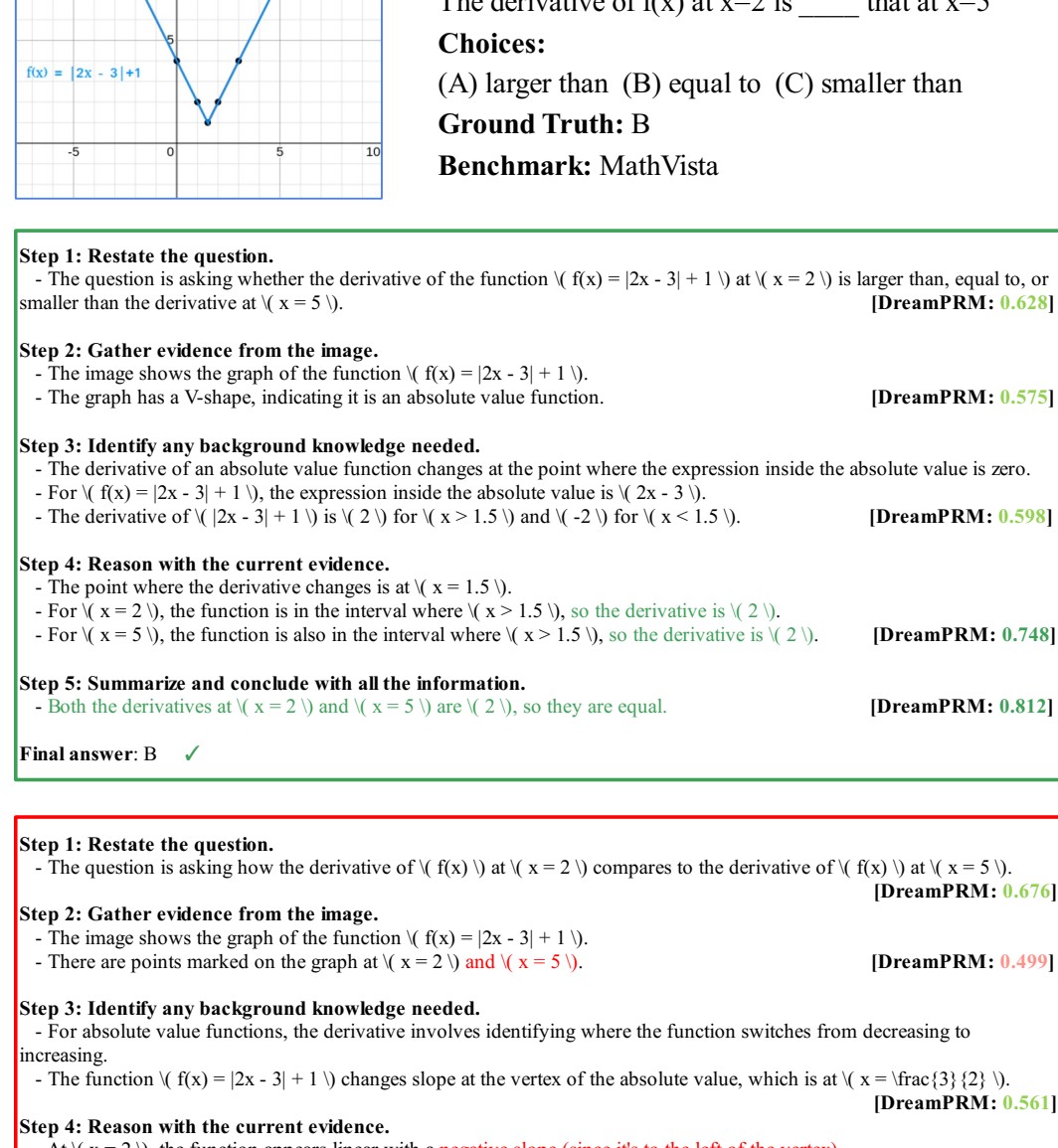

**Question:**

The derivative of f(x) at x=2 is _____ that at x=5

**Choices:**

(A) larger than  (B) equal to  (C) smaller than

**Ground Truth:** B

**Benchmark:** MathVista

---

**Step 1: Restate the question.**
  - The question is asking whether the derivative of the function $f(x) = |2x - 3| + 1$ at $x = 2$ is larger than, equal to, or smaller than the derivative at $x = 5$. **[DreamPRM: 0.628]**

**Step 2: Gather evidence from the image.**
  - The image shows the graph of the function $f(x) = |2x - 3| + 1$.
  - The graph has a V-shape, indicating it is an absolute value function. **[DreamPRM: 0.575]**

**Step 3: Identify any background knowledge needed.**
  - The derivative of an absolute value function changes at the point where the expression inside the absolute value is zero.
  - For $f(x) = |2x - 3| + 1$, the expression inside the absolute value is $2x - 3$.
  - The derivative of $|2x - 3| + 1$ is $2$ for $x > 1.5$ and $-2$ for $x < 1.5$. **[DreamPRM: 0.598]**

**Step 4: Reason with the current evidence.**
  - The point where the derivative changes is at $x = 1.5$.
  - For $x = 2$, the function is in the interval where $x > 1.5$, so the derivative is $2$.
  - For $x = 5$, the function is also in the interval where $x > 1.5$, so the derivative is $2$. **[DreamPRM: 0.748]**

**Step 5: Summarize and conclude with all the information.**
  - Both the derivatives at $x = 2$ and $x = 5$ are $2$, so they are equal. **[DreamPRM: 0.812]**

**Final answer**: B  ✓

---

**Step 1: Restate the question.**
  - The question is asking how the derivative of $f(x)$ at $x = 2$ compares to the derivative of $f(x)$ at $x = 5$. **[DreamPRM: 0.676]**

**Step 2: Gather evidence from the image.**
  - The image shows the graph of the function $f(x) = |2x - 3| + 1$.
  - There are points marked on the graph at $x = 2$ and $x = 5$. **[DreamPRM: 0.499]**

**Step 3: Identify any background knowledge needed.**
  - For absolute value functions, the derivative involves identifying where the function switches from decreasing to increasing.
  - The function $f(x) = |2x - 3| + 1$ changes slope at the vertex of the absolute value, which is at $x = \frac{3}{2}$. **[DreamPRM: 0.561]**

**Step 4: Reason with the current evidence.**
  - At $x = 2$, the function appears linear with a negative slope (since it's to the left of the vertex).
  - At $x = 5$, the function appears linear with a positive slope (since it's to the right of the vertex).
  - The slope at $x = 2$ (negative) is steeper/more negative than the slope at $x = 5$ (positive). **[DreamPRM: 0.397]**

**Step 5: Summarize and conclude with all the information.**
  - The derivative at $x = 2$ is negative and steeper than the derivative at $x = 5$, which is positive.
  - Therefore, the derivative at $x = 2$ is larger in absolute value than the derivative at $x = 5$. **[DreamPRM: 0.396]**

**Final answer:** A  ✗

Figure 10: A case study of DreamPRM's step-wise evaluation.

