# OpenReview forum: "DreamPRM: Domain-reweighted Process Reward Model for Multimodal Reasoning"
_NeurIPS.cc/2025/Conference — NeurIPS 2025 poster_

### Official Review · Reviewer_S8kQ · 2025-07-01

**Clarity:** 4
**Significance:** 3
**Originality:** 3
**Rating:** 4
**Confidence:** 3

**Summary:**

The paper propose a new domain-reweighted PRM framework for multimodal reasoning. They define it as a bi-level optimization framework, and optimize the PRM parameter in the lower level and domain weights in the upper level by turn. Extensive experiments demonstrate the effectiveness of their method compared to other data selection strategy.

**Questions:**

1. How do you design the aggregation function in Equation 2? Why using this form?

2. Your domain-reweighted method may be novel in MLLM PRM reasoning. Are there similar methods in other fields, such as LLM reasoning?

3. How do you select data from MMLU to build the meta dataset in the upper level optimization? This step is crucial in order to fully find the training imbalance across different domains.

**Ethical Concerns:**

["NO or VERY MINOR ethics concerns only"]

**Final Justification:**

I have read the rebuttal and the comments of the other reviewers. My final rating is borderline accept.

**Limitations:**

Their method assumes a fixed set of domains and requires Monte-Carlo sampling, which can be computationally heavy.

They use an approximated algorithm to solve the bi-level optimization problem, which may lead to unstable training and sub-optimal results.

**Paper Formatting Concerns:**

None major formatting issues.

**Quality:**

3

**Strengths And Weaknesses:**

**Strengths**:

Good Novelty. The paper regard the PRM test-time scaling optimization as a bi-level optimization framework, and design the corresponding method.

Clear writing. The paper is well organized and easy to understand.

**Weaknesses**:

As directly solving the bi-level optimization problem can be computational prohibitive, they use an approximated algorithm instead.However, this may lead to unstable training and sub-optimal results.

The meta dataset used in upper-level optimization may need to be carefully designed, in order to fully find the training imbalance across different domains.

---

> ### Author Rebuttal · Authors · 2025-07-30
>
> ## Common Response to All Reviewers
> We would like to thank all the reviewers for the constructive and valuable feedback, which we will leverage to improve this paper. We are encouraged by the positive comments from reviewers, including:
>
> - **Method.**
>   “a novel domain-reweighted training framework” (Reviewer Cvdg),
>   “well motivated” (Reviewer RhCp),
>   “Good Novelty.” (Reviewer S8kQ),
>   “novel in MLLM PRM reasoning” (Reviewer S8kQ).
>
> - **Experimental results.**
>   “baselines are strong” (Reviewer Cvdg),
>   “Extensive experiments demonstrate the effectiveness of their method” (Reviewer S8kQ).
>
> - **Writing.**
>   “well-written and easy to follow” (Reviewer RhCp),
>   “visualization is clear and comprehensive.” (Reviewer RhCp),
>   “Clear writing” (Reviewer S8kQ),
>   “well organized and easy to understand” (Reviewer S8kQ).
>
> ---
> **New Experiment: DreamPRM Achieves 1st Place on MathVista Leaderboard**
>
> To further strengthen our submission, we conducted a new evaluation on the MathVista benchmark. In this experiment, we applied our proposed DreamPRM on top of the stronger o4-mini model, under an 8-CoT Best-of-N (BoN) setting.
>
> Notably, this result ranks **1st on the official MathVista leaderboard**, further confirming the state-of-the-art performance of our method. A snapshot of the top leaderboard entries is provided below:
> **Top 5 on the MathVista Leaderboard (as of submission)**
>
> | Rank | Model                        | Date       | Accuracy (%) |
> |------|------------------------------|------------|---------------|
> | 1    | **DreamPRM (o4-mini)**       | 2025-06-04 | **85.2**      |
> | 2    | VL-Rethinker                 | 2025-04-10 | 80.3          |
> | 3    | Step R1-V-Mini               | 2025-04-07 | 80.1          |
> | 4    | Kimi-k1.6-preview-20250308   | 2025-03-10 | 80.0          |
> | 5    | Doubao-pro-1.5               | 2025-01-22 | 79.5          |
>
> This newly added experiment demonstrates the effectiveness and generalizability of DreamPRM: our method achieves **85.2% accuracy**, outperforming both the base model (80.6%) and a self-consistency baseline (81.7%).
>
> | Method                              | MathVista Accuracy (%) |
> |-------------------------------------|------------------------:|
> | DreamPRM + o4-mini (8 CoTs)         | **85.2**                |
> | Self-consistency + o4-mini (8 CoTs) | 81.7                    |
> | o4-mini (pass@1)                    | 80.6                    |
>
> The responses to each reviewer's comments are provided below.
>
> ---
> ## Response to Reviewer S8kQ
> We are grateful to the reviewer for the positive and constructive review. New results and discussions will be added to the revised version of this paper.
>
> ***Response to the Design Choice of the Aggregation Function:***
>
> **1. How do we design the aggregation function**
>
> We adopt the aggregation function proposed in the MiPS paper [1], which recommends using functions that emphasize high scores when aggregating Monte Carlo annotations. Specifically, the MiPS study shows that **aggregation functions which value high scores highly**—such as `max` or our used aggregation function—are most effective for this type of data. Since our PRM is trained on Monte Carlo–annotated data, this design choice is both theoretically justified and well-aligned with prior findings.
>
> We follow the same intuition: *a good aggregation function should give more weight to high scores*.
>
> **2. Why we use this form in practice**
>
> In our upper-level training, we found that simple alternatives such as the mean aggregation significantly degrade performance. As shown in our ablation study (Figure 3c and Appendix Table 4), replacing the aggregation function loss with mean aggregation function loss results in worse performance. This suggests that a carefully designed aggregation function is necessary for stable and effective training.
>
> We further validated this design at inference time by comparing different aggregation functions on the MathVista benchmark:
>
> | Aggregation Method        | Accuracy (%) |
> |---------------------------|---------------|
> | Min                       | 68.1          |
> | Max                       | 68.2          |
> | Mean                      | 67.8          |
> | Our aggregation function  | **68.9**      |
>
> These results confirm that both `max` and our aggregation function outperform `min` and `mean`, which supports the MiPS paper’s observation that functions emphasizing high scores perform better. However, compared to `max`, our aggregation function is differentiable and better suited for gradient-based training in the upper level.
>
> Hence, we adopt the MiPS-style aggregation function for both theoretical motivation and practical performance.
>
> ---
> ***Response to Question on Domain Reweighting in Other Fields***
>
> To the best of our knowledge, our work is the first to introduce domain reweighting for reasoning tasks, particularly in the context of multimodal process reward models (PRMs).
>
> There is no prior work that applies domain reweighting to LLM reasoning. Related methods, such as DoGE [2], focus on using domain reweighting for improving generalization during LLM pretraining.
>
> For more details, please refer to Appendix A: Domain Reweighting.
>
> ---
> ***Response to Question on Meta Dataset Selection from MMMU***
>
> **1. How do we select the meta dataset**
>
> We construct the meta dataset by selecting samples from MMMU where the base model generates both correct and incorrect CoTs under the Best-of-N (BoN) setting. For each such question, we include both the correct and incorrect CoTs. This setup encourages the PRM to learn to rank correct reasoning paths higher than incorrect ones, which matches its usage during inference.
>
> To validate this selection strategy, we conduct an ablation on MathVista using InternVL:
>
> | Meta Dataset Type                  | MathVista Accuracy (%) |
> |------------------------------------|-------------------------|
> | All positive samples               | 66.4                    |
> | Positive + negative samples (ours) | **68.9**                |
>
> **2. Concern on meta dataset design**
>
> We acknowledge the reviewer’s concern, but note that the meta dataset design in our framework is not complex. The only requirement is that the base model generates both correct and incorrect responses, which naturally occurs in practice.
>
> ---
> ***Response to Comment on Bi-level Optimization Stability:***
>
> We thank the reviewer for pointing out that approximate bilevel solvers can suffer from instability and sub-optimal convergence. We address this in two parts:
>
> **1. Theoretical justification via MAML**
>
> Meta-learning methods such as MAML [3] also rely on a truncated, “approximate” inner loop when solving the bi-level problem – yet they achieve stable and effective adaptation in practice. This demonstrates that approximation alone does not preclude convergence, provided the algorithm and hyperparameters are well-chosen.
>
> **2. Our practical implementation using Betty**
>
> - **Best-response Jacobian:** We implement our bi-level updates using the Betty [4] library, which incorporates the *best-response Jacobian* technique to more accurately approximate the gradient of the upper‐level loss with respect to domain weights. This approach corrects for bias introduced by truncation and yields more reliable updates.
> - **Empirical convergence:** As shown in Appendix E (Figure 8), we plot both (a) the bilevel training loss and (b) the evolution of the domain weights over training iterations. These curves demonstrate smooth, monotonic convergence and stable weight assignments across domains, confirming that our solver does not suffer from erratic behavior.
>
> ---
> ***Response to Concern on the Efficiency of Monte Carlo Annotation***
>
> We acknowledge that Monte Carlo–based annotation can be time-consuming, but it remains one of the most widely used and effective strategies for constructing step-level reasoning supervision—especially in multimodal tasks where high-quality human annotations are scarce. For instance, Math-Shepherd [5] and VisualPRM [6] also adopt this approach.
>
> In our case, Monte Carlo sampling is used to generate human-free supervision signals for intermediate reasoning steps. As detailed in Appendix C, we further improve the efficiency of this process by introducing a **dynamic sampling strategy**:
>
> > “We propose a similar dynamic sampling technique for Monte Carlo estimation that focuses on prompts with varied outcomes to improve efficiency.”
>
> This method prioritizes samples with high variance, which helps reduce the number of samples required to obtain a meaningful training signal.
>
> We also note that this annotation process is conducted once offline, so while it incurs an upfront cost, it does not affect efficiency. Furthermore, to benefit the community, we plan to release our training set, which we hope will facilitate future research in multimodal reasoning.
>
> ---
> **References**
> [1] Wang et al. *Multi-step problem solving through a verifier: An empirical analysis on model-induced process supervision.* arXiv preprint arXiv:2402.02658 (2024).
> [2] Fan et al. *Doge: Domain reweighting with generalization estimation.* arXiv preprint arXiv:2310.15393 (2023).
> [3] Finn et al. *Model-agnostic meta-learning for fast adaptation of deep networks.* In International conference on machine learning (pp. 1126-1135). PMLR.
> [4] Choe et al. *Betty: An automatic differentiation library for multilevel optimization.* arXiv preprint arXiv:2207.02849 (2022).
> [5] Wang et al. *Math-shepherd: Verify and reinforce llms step-by-step without human annotations.* arXiv preprint arXiv:2312.08935 (2023).
> [6] Wang et al.,*VisualPRM: An Effective Process Reward Model for Multimodal Reasoning*. arXiv:2503.10291 (2025).

---

> ### Author Response · Authors · 2025-08-04
>
> Dear Reviewer S8kQ,
>
> Thank you again for your valuable feedback and questions.
> We wonder whether our responses have sufficiently addressed your comments?
> If more information is needed, we are very happy to provide it.
> We greatly appreciate your time and effort in this process.
>
> Best,
> Authors

---

> > ### Comment · Reviewer_S8kQ · 2025-08-05
> >
> > Thank you for your response. I no longer have any further questions. I  will maintain my original score.

---

> ### Author Response · Authors · 2025-08-05
>
> We truly appreciate your time and constructive feedback. Thank you for carefully reviewing our response.

---

### Official Review · Reviewer_RhCp · 2025-07-03

**Clarity:** 3
**Significance:** 2
**Originality:** 2
**Rating:** 4
**Confidence:** 4

**Summary:**

The paper introduces a domain-reweighted PRM for multimodal reasoning tasks. The authors propose to train the PRM with domain weights in the lower level while training the domain weights on a meta dataset on the upper level. This is the Bi-level technique widely used in meta-learning. The authors conduct extensive experiments on multiple benchmarks.

**Questions:**

1. Does the proposed training method affect the inference efficiency?
2. How can $\alpha$ be updated by the meta loss? And what is the motivation for the design of the form of meta loss?
3. Test science benchmarks and filtered MathVista for more reliable results.
I will raise my score if the concerns in the questions and weaknesses parts are addressed.

**Ethical Concerns:**

["NO or VERY MINOR ethics concerns only"]

**Final Justification:**

The author has resolved my concerns in the rebuttal, mostly regarding the data leakage issue. Overall, the paper is well-motivated with apparent improvements on different benchmarks.

**Limitations:**

yes

**Paper Formatting Concerns:**

No major formatting issues spotted.

**Quality:**

3

**Strengths And Weaknesses:**

Strengths:
1. The paper is well motivated. The authors identify the low-quality data when training the multimodal PRM and propose a sensible method by assigning different weights.
2. The paper is well-written and easy to follow. The visualization is clear and comprehensive.

Weaknesses:
1. The major concern is the data leakage. The training dataset contains several benchmarks that also exist in MathVista, such as geos, geometry3k, iconQA, and ai2d. So the results of MathVista are not reliable. The authors need to eliminate the data in these benchmarks and examine if there is any other data leakage in MathVista. The authors should test on this filtered version and report the results.
2. The tested benchmarks are limited. The training dataset contains many benchmarks in the science domain. The authors should consider evaluating the proposed method on science benchmarks like SciVerse [1] and OlympiadBench [2]. The evaluation result of OlympiadBench could also help to verify the generalization capability of the proposed DreamPRM on difficult tasks.
3. The upper-level optimization is confusing. How is the meta loss related to $\alpha$? Why are there gradients on $\alpha$ in the meta loss? The motivation of the meta loss is also unclear. How does this more reflect the inference process?

[1] Guo, Z., Zhang, R., Chen, H., Gao, J., Jiang, D., Wang, J., & Heng, P. A. (2025). Sciverse: Unveiling the knowledge comprehension and visual reasoning of lmms on multi-modal scientific problems. arXiv preprint arXiv:2503.10627.

[2] He, C., Luo, R., Bai, Y., Hu, S., Thai, Z. L., Shen, J., ... & Sun, M. (2024). Olympiadbench: A challenging benchmark for promoting agi with olympiad-level bilingual multimodal scientific problems. arXiv preprint arXiv:2402.14008.

---

> ### Author Rebuttal · Authors · 2025-07-30
>
> ## Common Response to All Reviewers
> We would like to thank all the reviewers for the constructive and valuable feedback, which we will leverage to improve this paper. We are encouraged by the positive comments from reviewers, including:
>
> - **Method.**
>   “a novel domain-reweighted training framework” (Reviewer Cvdg),
>   “well motivated” (Reviewer RhCp),
>   “Good Novelty.” (Reviewer S8kQ),
>   “novel in MLLM PRM reasoning” (Reviewer S8kQ).
>
> - **Experimental results.**
>   “baselines are strong” (Reviewer Cvdg),
>   “Extensive experiments demonstrate the effectiveness of their method” (Reviewer S8kQ).
>
> - **Writing.**
>   “well-written and easy to follow” (Reviewer RhCp),
>   “visualization is clear and comprehensive.” (Reviewer RhCp),
>   “Clear writing” (Reviewer S8kQ),
>   “well organized and easy to understand” (Reviewer S8kQ).
>
> ---
> **New Experiment: DreamPRM Achieves 1st Place on MathVista Leaderboard**
>
> To further strengthen our submission, we conducted a new evaluation on the MathVista benchmark. In this experiment, we applied our proposed DreamPRM on top of the stronger o4-mini model, under an 8-CoT Best-of-N (BoN) setting.
>
> Notably, this result ranks **1st on the official MathVista leaderboard**, further confirming the state-of-the-art performance of our method. A snapshot of the top leaderboard entries is provided below:
> **Top 5 on the MathVista Leaderboard (as of submission)**
>
> | Rank | Model                        | Date       | Accuracy (%) |
> |------|------------------------------|------------|---------------|
> | 1    | **DreamPRM (o4-mini)**       | 2025-06-04 | **85.2**      |
> | 2    | VL-Rethinker                 | 2025-04-10 | 80.3          |
> | 3    | Step R1-V-Mini               | 2025-04-07 | 80.1          |
> | 4    | Kimi-k1.6-preview-20250308   | 2025-03-10 | 80.0          |
> | 5    | Doubao-pro-1.5               | 2025-01-22 | 79.5          |
>
> This newly added experiment demonstrates the effectiveness and generalizability of DreamPRM: our method achieves **85.2% accuracy**, outperforming both the base model (80.6%) and a self-consistency baseline (81.7%).
>
> | Method                              | MathVista Accuracy (%) |
> |-------------------------------------|------------------------:|
> | DreamPRM + o4-mini (8 CoTs)         | **85.2**                |
> | Self-consistency + o4-mini (8 CoTs) | 81.7                    |
> | o4-mini (pass@1)                    | 80.6                    |
>
> The responses to each reviewer's comments are provided below.
>
> ---
> ## Response to Reviewer RhCp
> We are grateful to the reviewer for the constructive review. New results and discussions will be added to the revised version of this paper.
>
> ***Response to Concern on Potential Data Leakage in Evaluation***
>
> We thank the reviewer for raising this important point. We are confident that there is **no data leakage** between our training data and any evaluation benchmark, including MathVista. We employ two strong safeguards:
>
> **1. Subsampling with Filtering:**
> As detailed in Appendix C, we do not use full datasets. Instead, we sample ~1,000 representative examples per domain (15 domains total, ~15k samples). To prevent overlap with test sets, we apply three filters:
>
> - `is_duplicate_match`: removes samples that are identical to any test instance
> - `is_partial_match`: removes samples with partial overlap on key text segments
> - `is_similarity_match`: uses `difflib.SequenceMatcher` with a 0.7 threshold to remove semantically similar cases
>
> These filters effectively ensure no data leakage.
>
> **2. Use of Publicly Released, Filtered Data:**
> Our data comes from the MMPR [1] dataset used by InternVL [2], a fully open-source model. We use only the question texts (not responses), and apply our own filtering. Since MMPR is independently curated and public, this ensures no data leakage again.
>
> We will also release our filtered training splits for transparency.
>
> ---
>
> ***Response to Concern on Limited Benchmark Scope and Suggestion to Evaluate on Science Benchmarks***
>
> We thank the reviewer for the suggestion to include more science-focused benchmarks such as SciVerse and OlympiadBench. In response, we have conducted additional evaluations on both benchmarks, using two base models—InternVL-2.5-8B-MPO and o4-mini—and compared DreamPRM with three standard baselines: No selection, Self-Correction and Self-Consistency.
>
> For SciVerse, each question has five versions. In our evaluation, we use the **knowledge-free** version, which does not rely on external retrieval and better reflects a model’s intrinsic reasoning capability.
>
> For OlympiadBench, the full dataset consists of 18 subsets, including multilingual and purely textual variants. To ensure a fair and focused evaluation aligned with our multimodal setup, we report results on the **OE_MM_maths_en_COMP** subset, which contains English multimodal math problems with image and text components.
>
> We acknowledge that OlympiadBench is particularly challenging, especially for smaller models like InternVL. Therefore, we additionally evaluate with o4-mini, a stronger base model.
>
> **Results on InternVL-2.5-8B-MPO:**
>
> | Model               | SciVerse | OlympiadBench |
> |---------------------|----------|---------------|
> | InternVL-2.5-8B-MPO | 47.4     | 10.2          |
> | No selection        | 47.8     | 11.3          |
> | Self-Correction     | 45.6     | 11.3          |
> | Self-Consistency    | 48.9     | 10.7          |
> | DreamPRM (Ours)     | **49.3** | **13.4**      |
>
> **Results on o4-mini:**
>
> | Model             | OlympiadBench |
> |------------------|---------------|
> | o4-mini           | 85.9          |
> | No selection      | 87.9          |
> | Self-Correction   | 87.9          |
> | Self-Consistency  | 89.9          |
> | DreamPRM (Ours)   | **90.6**      |
>
> These results confirm that DreamPRM significantly improves performance over standard baselines, particularly on difficult science reasoning tasks. Notably, it achieves state-of-the-art performance on OlympiadBench when paired with o4-mini, demonstrating its strong generalization capability.
>
> ---
> ***Response to Concern on Upper-Level Optimization and the Inference Process***
>
> We thank the reviewer for the insightful question. We clarify the role of the meta loss and its connection to $\alpha$ and the inference process in 3 parts:
>
> **1. How is the meta loss related to $\alpha$**
>
> In the bi-level optimization setup, the lower-level objective is:
>
> $$
> \mathcal{L}\_{\text{tr}}(\mathcal{D}\_{\text{tr}}, \phi, \alpha) = \sum_ {k=1}^{K} \alpha\_k \mathcal{L}\_{\text{tr}}(\mathcal{D}\_k, \phi)
> $$
>
> We optimize $\phi$ while holding $\alpha$ fixed, and denote the optimal solution as:
>
> $$
> \phi^*(\alpha) = \arg\min\_{\phi} \mathcal{L}\_{\text{tr}}(\mathcal{D}\_{\text{tr}}, \phi, \alpha)
> $$
>
> This makes $\phi^* $ a function of $\alpha$. In the upper level, the meta loss is defined using $\phi^* (\alpha)$:
>
> $$
> \mathcal{L}\_{\text{meta}}(\mathcal{D}\_{\text{meta}}, \phi^* (\alpha)) = \sum_ {(x, y) \in \mathcal{D}\_{\text{meta}}} \mathcal{L}\_{\text{MSE}} \left( \sigma(\mathcal{A}(\mathcal{V}\_{\phi^* (\alpha)}(x, \hat{y}))),\ r(\hat{y}, y) \right)
> $$
>
> By function composition, it is clear that the meta loss is a function of $\alpha$, i.e., $\mathcal{L}\_{\text{meta}} = \mathcal{F}(\phi^* (\alpha))$, and thus $\nabla\_\alpha \mathcal{L}\_{\text{meta}}$ exists.
>
> **2. How to compute the gradient of $\alpha$ in practice**
>
> To compute $\nabla_\alpha \mathcal{L}_{\text{meta}}$, we adopt a one-step gradient descent approximation for $\phi^*(\alpha)$ following prior work such as DARTS [3]:
>
> $$
> \phi^{(t+1)} = \phi^{(t)} - \beta\_1 \nabla\_\phi \mathcal{L}\_{\text{tr}}(\mathcal{D}\_{\text{tr}}, \phi, \alpha)
> $$
>
> This yields an approximate form:
>
> $$
> \phi^* (\alpha) \approx \phi^{(t)} - \beta\_1 \nabla\_\phi \mathcal{L}\_{\text{tr}}(\mathcal{D}\_{\text{tr}}, \phi) \cdot \alpha
> $$
>
> Differentiating the meta loss w.r.t. $\alpha$ follows the chain rule:
>
> $$
> \frac{\partial \mathcal{L}\_{\text{meta}}}{\partial \alpha} = \frac{\partial \mathcal{L}\_{\text{meta}}}{\partial \phi^* (\alpha)} \cdot \frac{\partial \phi^* (\alpha)}{\partial \alpha}
> $$
>
> and from the approximate form:
>
> $$
> \frac{\partial \phi^* (\alpha)}{\partial \alpha} = \frac{\partial}{\partial \alpha} \left( \phi^{(t)} - \beta_1 \nabla\_\phi \mathcal{L}\_{\text{tr}}(\mathcal{D}\_{\text{tr}}, \phi) \cdot \alpha \right)
> $$
>
> This gives a practical way to compute gradients on $\alpha$ for upper-level optimization.
>
> **3. Why meta loss better reflects inference**
>
> At inference, PRM selects the best CoT from $N$ candidates by:
>
> $$
> \text{score} = \mathcal{A}(\mathcal{V}_{\phi}(x, \hat{y}))
> $$
>
> where $\mathcal{A}$ is the aggregation function, $x$ is the input, and $\hat{y}$ is a candidate CoT. This is identical to the meta loss formulation. During training, we apply a sigmoid and encode correctness as a binary label $r(\hat{y}, y) \in {0, 1}$, with 1 for correct CoTs and 0 otherwise. While lower-level loss supervises step-level predictions, the upper-level loss evaluates the **whole reasoning trail**, which matches inference.
>
> ---
> ***Response to Concern on Inference Efficiency***
>
> Our proposed method introduces a new training framework for the PRM, but **does not affect inference efficiency**. At inference time, the PRM is used in the same way as in prior methods—evaluating and aggregating scores over candidate reasoning steps (e.g., in a Best-of-N setting). The inference pipeline remains unchanged, with no additional computational overhead introduced by our training procedure.
>
> ---
> **References**
> [1] Wang et al. *Enhancing the reasoning ability of multimodal large language models via mixed preference optimization.* arXiv preprint (2024).
> [2] Chen et al. *Internvl: Scaling up vision foundation models and aligning for generic visual-linguistic tasks.* Proceedings of the IEEE/CVF conference on computer vision and pattern recognition. (2024).
> [3] Liu et al. "Darts: Differentiable architecture search." arXiv preprint (2018).

---

> ### Comment · Reviewer_RhCp · 2025-08-01
>
> Thanks for the authors' response. Most of my concerns are addressed. However, I remain confused about several points:
>
> 1. Regarding potential data leakage: The authors claim they "only used the question texts instead of responses." However, PRM training requires ground truth answers for supervision. So the model has still seen the ground truth during training. How does using only question texts address the data leakage concern?
>
> 2. Regarding the MMPR dataset: The authors claim MMPR is "independently curated and public," but I cannot find any statement in the MMPR paper about dataset filtering. Could the authors clarify where this filtering process is documented?
>
> 3. Regarding the test of o4-mini. Does the response of o4-mini contain the full CoT process or a summarized one? Could the authors provide some examples of the o4-mini response and the score of the PRM?

---

> > ### Author Response · Authors · 2025-08-01
> > **Response to Reviewer RhCp – Questions 1 & 2 (Question 3 addressed in the following comment due to character limit)**
> >
> > ## Response to Reviewer RhCp （Part 1/2）
> > **Question 1**
> > Thanks for pointing out this concern. Our statement "only used question texts instead of responses" was intended to clarify how we utilized MMPR data. Originally, MMPR contains long CoT responses generated by large language models, which we explicitly did not use. Instead, we generated our own responses for Monte Carlo estimation, as MMPR lacks the step-wise annotation necessary for PRM training.
> >
> > We acknowledge that generating these step-wise annotations naturally involves the ground truth answers. Thus, our wording aimed primarily to emphasize that we did not use MMPR’s provided responses (CoTs), not that this step alone prevents data leakage. To address potential data leakage explicitly, we applied the three data filters mentioned in our response.
> >
> > ---
> > **Question 2**
> > We apologize for the unclear explanation and any confusion caused in our previous response.
> >
> > Our mention of the MMPR dataset is primarily intended to **ensure transparency of our data processing pipeline**. Since MMPR is a publicly available dataset that spans multiple domains, it allows us to more clearly demonstrate how we construct our training data. By explicitly referencing MMPR, we aim to help reviewers better understand each step of our processing, rather than suggesting MMPR inherently prevents data leakage.
> >
> > To clarify, we did **not** use the long CoT responses provided in MMPR. Instead, we only used the question–answer pairs from MMPR’s `vqa_correctness_rules` subset. This subset was selected because the questions have **clearly defined and verifiable answers**, which are essential for accurate Monte Carlo estimation in our PRM training.
> >
> > Indeed, MMPR itself applied filtering for `vqa_correctness_rules` subset, as described in its paper [1]:
> >
> > > “Through the correctness-based pipeline, we exclude questions from general VQA and document sources, as verifying the correctness of the generated answers using heuristic rules is challenging for datasets in these domains. For example, the ground truths in VQAv2 consist of a single word or phrase, which may lead to false-negative responses when the model outputs a complete sentence or a synonym as the final answer. Such false-negative responses can negatively impact training effectiveness.”
> >
> > However, we **fully recognized that such filtering in MMPR does not fully prevent potential data leakage**. Therefore, we further applied our own filtering procedures to remove any samples that overlap or closely resemble instances in our evaluation benchmarks.
> >
> > To clearly illustrate our data construction pipeline, we summarize it below:
> >
> > MMPR question-answer pairs (from multiple domains)
> > → MMPR’s vqa_correctness_rules subset (facilitates Monte Carlo)
> > → Our filtering methods (prevents data leakage)
> > → Monte Carlo annotation
> > → Our final training set
> >
> >
> > We emphasize again that our goal in mentioning MMPR is **not to claim leakage is avoided by using a public dataset**, but rather to make our data processing pipeline **as transparent as possible**. We hope this clearer explanation addresses your concerns.
> >
> > (*Question 3 is addressed in the following comment due to character limit*)
> >
> > ---
> > **References**
> > [1] Wang et al. Enhancing the reasoning ability of multimodal large language models via mixed preference optimization. arXiv preprint (2024).

---

> > > ### Author Response · Authors · 2025-08-01
> > > **Response to Reviewer RhCp – Question 3**
> > >
> > > ## Response to Reviewer RhCp （Part 2/2）
> > >
> > > **Question 3**
> > > We used the `output_text` from o4-mini response, and did not use the `summary`. Below, we provide an example illustrating the o4-mini responses along with their corresponding PRM scores.
> > >
> > > *Question 21 from OlympiadBench*
> > >
> > > o4-mini's correct response:
> > >
> > > >Step 1: Restate the question.
> > > We have rectangle \(ABCD\).  Point \(E\) on side \(AB\) is such that \(\angle AEB=90^\circ\), \(AE=15\), \(EB=20\).  Point \(F\) on side \(DC\) is such that \(\angle DFC=90^\circ\) and \(DF=24\).  We are asked to find the length \(CF\).
> > > >
> > > >Step 2: Gather evidence from the images.
> > > – Triangle \(AEB\) is drawn inside the rectangle with a right angle at \(E\).  The legs are labeled \(AE=15\) and \(EB=20\).
> > > – Triangle \(DFC\) is drawn inside with a right angle at \(F\).  One leg is \(DF=24\), the other leg \(FC\) is unlabeled, and the hypotenuse \(DC\) is the side of the rectangle.
> > > >
> > > >Step 3: Identify any background knowledge needed.
> > > – In a right triangle, the Pythagorean theorem holds: hypotenuse\(^2 =\) (leg\(_1\))\(^2 +\)(leg\(_2\))\(^2\).
> > > – Opposite sides of a rectangle are equal in length, so \(AB=DC\).
> > > >
> > > >Step 4: Reason with the current evidence.
> > > – In right triangle \(AEB\), the hypotenuse is \(AB\), so
> > >   \(AB = \sqrt{AE^2 + EB^2} = \sqrt{15^2 + 20^2} = \sqrt{225 + 400} = \sqrt{625} = 25.\)
> > >   Thus \(AB=25\), and hence \(DC=25\).
> > > – In right triangle \(DFC\), the hypotenuse is \(DC=25\) and one leg is \(DF=24\).  By the Pythagorean theorem,
> > >   \(CF = \sqrt{DC^2 - DF^2} = \sqrt{25^2 - 24^2} = \sqrt{625 - 576} = \sqrt{49} = 7.\)
> > > >
> > > >Step 5: Summarize and conclude with all the information.
> > > Using the two inscribed right triangles and the fact that \(AB=DC\), we find \(AB=25\) and then compute \(CF=\sqrt{25^2-24^2}=7\).
> > > >
> > > >Final answer: 7
> > >
> > > PRM scores: [0.589, 0.441, 0.429, 0.438, 0.506]
> > >
> > > o4-mini's incorrect  response:
> > > >Step 1: Restate the question.
> > > We have rectangle ABCD. Inside it two “broken” lines are drawn: A–E–B, with AE=15, EB=20 and a right angle at E; and D–F–C, with DF=24, FC unknown and a right angle at F. We are asked to find the length CF.
> > > >
> > > >Step 2: Gather evidence from the image.
> > > >- ABCD is a rectangle, so AB∥DC and AD∥BC.
> > > >- On the interior, A–E–B is drawn with ∠AEB=90°, AE=15, EB=20.
> > > >- Also D–F–C is drawn with ∠DFC=90°, DF=24, FC is unlabeled.
> > > >- Visually, the segment AE slopes down‐right, and FC slopes down‐right by the same angle; EB slopes up‐right, and DF slopes up‐right by the same angle.
> > > >
> > > >Step 3: Identify any background knowledge needed.
> > > >- In a rectangle opposite sides are parallel.
> > > >- If two angles are right angles and the legs of one right triangle are parallel respectively to the legs of another right triangle, the triangles are similar.
> > > >- In similar triangles corresponding sides are proportional.
> > > >
> > > >Step 4: Reason with the current evidence.
> > > >- Since AD∥BC∥AE∥FC and AB∥DC∥EB∥DF, the right triangle AEB is similar to right triangle DFC by AA (right angle + one acute angle).
> > > >- Corresponding legs: AE↔FC and EB↔DF.
> > > >- Therefore FC/DF = AE/EB.
> > > >
> > > >Step 5: Summarize and conclude with all the information.
> > > >From similarity, FC = DF·(AE/EB) = 24·(15/20) = 24·0.75 = 18.
> > > >
> > > >Final answer: 18
> > >
> > > PRM scores: [0.560, 0.370, 0.229, 0.166, 0.271]
> > >
> > > We can see that the scores of o4-mini's incorrect response is much lower than the correct one.

---

> > > > ### Comment · Reviewer_RhCp · 2025-08-01
> > > >
> > > > Thanks for the author's swift response. My concerns are adequately resolved, and I have raised my score to 4.

---

> > > > > ### Author Response · Authors · 2025-08-01
> > > > >
> > > > > We truly appreciate your time and constructive feedback. Thank you for carefully reviewing our response and for raising the score — it means a lot to us.

---

### Official Review · Reviewer_Cvdg · 2025-07-05

**Clarity:** 2
**Significance:** 2
**Originality:** 3
**Rating:** 4
**Confidence:** 2

**Summary:**

This paper introduces DreamPRM, a novel domain-reweighted training framework for multimodal Process Reward Models (PRMs) designed to tackle the dataset quality imbalance and distribution shift challenges in multimodal reasoning. The core of the method is a bi-level optimization (BLO) framework that dynamically learns weights for different training domains. The lower level optimizes the PRM on reweighted data, while the upper level updates these domain weights using a novel aggregation function loss on a high-quality meta-dataset, effectively aligning the training objective with the inference-time evaluation.

**Questions:**

N/A

**Ethical Concerns:**

["NO or VERY MINOR ethics concerns only"]

**Final Justification:**

rasing score from 3 to 4 based on additinal information during rebuttal

**Limitations:**

yes

**Quality:**

2

**Strengths And Weaknesses:**

Analysis of Test-Time Scaling: The performance gain from voting-based methods like Self-consistency in Table 1 appears marginal, which contrasts with the significant improvements often seen in text-only reasoning. It would be insightful to report the pass@8 accuracy to better understand the performance ceiling of the proposed approach.

Clarity of Figures and Analysis: The radar charts in Figures 4 and 5 are difficult to interpret due to the inconsistent scales across different dataset axes, potentially misrepresenting the magnitude of improvements. Furthermore, the sensitivity to the proposed method varies significantly across benchmarks (Figure4.c and Figure5）. A discussion explaining why certain datasets are more responsive than others would strengthen the analysis.

Inclusion of More Relevant Baselines: While the baselines are strong, the comparison could be more comprehensive. The authors are encouraged to include other recently developed Process Reward Models (PRMs) that are also specifically designed for multimodal reasoning to provide a more direct assessment of DreamPRM's novelty and performance.

---

> ### Author Rebuttal · Authors · 2025-07-30
>
> ## Common Response to All Reviewers
> We would like to thank all the reviewers for the constructive and valuable feedback, which we will leverage to improve this paper. We are encouraged by the positive comments from reviewers, including:
>
> - **Method.**
>   “a novel domain-reweighted training framework” (Reviewer Cvdg),
>   “well motivated” (Reviewer RhCp),
>   “Good Novelty.” (Reviewer S8kQ),
>   “novel in MLLM PRM reasoning” (Reviewer S8kQ).
>
> - **Experimental results.**
>   “baselines are strong” (Reviewer Cvdg),
>   “Extensive experiments demonstrate the effectiveness of their method” (Reviewer S8kQ).
>
> - **Writing.**
>   “well-written and easy to follow” (Reviewer RhCp),
>   “visualization is clear and comprehensive.” (Reviewer RhCp),
>   “Clear writing” (Reviewer S8kQ),
>   “well organized and easy to understand” (Reviewer S8kQ).
>
> ---
> **New Experiment: DreamPRM Achieves 1st Place on MathVista Leaderboard**
>
> To further strengthen our submission, we conducted a new evaluation on the MathVista benchmark. In this experiment, we applied our proposed DreamPRM on top of the stronger o4-mini model, under an 8-CoT Best-of-N (BoN) setting.
>
> Notably, this result ranks **1st on the official MathVista leaderboard**, further confirming the state-of-the-art performance of our method. A snapshot of the top leaderboard entries is provided below:
> **Top 5 on the MathVista Leaderboard (as of submission)**
>
> | Rank | Model                        | Date       | Accuracy (%) |
> |------|------------------------------|------------|---------------|
> | 1    | **DreamPRM (o4-mini)**       | 2025-06-04 | **85.2**      |
> | 2    | VL-Rethinker                 | 2025-04-10 | 80.3          |
> | 3    | Step R1-V-Mini               | 2025-04-07 | 80.1          |
> | 4    | Kimi-k1.6-preview-20250308   | 2025-03-10 | 80.0          |
> | 5    | Doubao-pro-1.5               | 2025-01-22 | 79.5          |
>
> This newly added experiment demonstrates the effectiveness and generalizability of DreamPRM: our method achieves **85.2% accuracy**, outperforming both the base model (80.6%) and a self-consistency baseline (81.7%).
>
> | Method                              | MathVista Accuracy (%) |
> |-------------------------------------|------------------------:|
> | DreamPRM + o4-mini (8 CoTs)         | **85.2**                |
> | Self-consistency + o4-mini (8 CoTs) | 81.7                    |
> | o4-mini (pass@1)                    | 80.6                    |
>
> The responses to each reviewer's comments are provided below.
>
> ---
> ## Response to Reviewer Cvdg
> We are grateful to the reviewer for the constructive review. New results and discussions will be added to the revised version of this paper.
>
> ***Response: Analysis of Test-Time Scaling***
>
> **1. On the modest gain from self-consistency (Table 1):**
> The modest improvement we observe aligns with the trend reported in Wang et al. (2022) [1], where self-consistency has better performance with larger models — e.g., +3% to +6% for UL2-20B and +9% to +23% for a 137B model. Thus, since all our experiments are conducted with 8B-scale models, which are relatively small, the self-consistency gains are expected to be modest.
>
> Moreover, we find that the effectiveness of self-consistency correlates with the distribution of correct CoTs (chains of thought). Benchmarks with a higher proportion of samples having 4–7 correct CoTs (out of 8) benefit more from self-consistency. For example:
>
> - **MathVision** has a high proportion of 0–3 correct CoTs → weak self-consistency gain
> - **WeMath** has a higher proportion of 4–7 correct CoTs → stronger self-consistency gain
>
> | Dataset       | % of Samples with 0 Correct CoTs | 1–3 Correct CoTs | 4–7 Correct CoTs | All 8 Correct | Self-Consistency Gain (%) |
> |---------------|----------------------------------|------------------|------------------|----------------|----------------------------|
> | MathVision    | 52.5%                           | 28.7%            | 13.9%            | 4.9%           | +0.3                       |
> | WeMath        | 21.6%                           | 20.7%            | 27.8%            | 29.9%          | +4.7                       |
>
> In contrast, PRM-based methods are more sensitive to the ratio between non-zero and zero correct CoTs (i.e., 1–7 vs. 0). For example, MathVista under InternVL shows limited gains under both PRM-based methods and self-consistency, potentially due to the low proportion of samples with 4–7 correct CoTs and a high proportion of samples with 0 correct CoTs. However, o4-mini yields a better distribution with a higher proportion of samples with 1-7 correct CoTs, making it more suitable for PRM:
>
> | Dataset (model)      | % of Samples with 0 Correct CoTs | 1–3 Correct CoTs | 4–7 Correct CoTs | All 8 Correct |
> |----------------------|---------|-----|-----|-----|
> | MathVista (InternVL) | 22.6%   | 9.1%  | 12.7% | 55.6% |
> | MathVista (o4-mini)  | 8.4%    | 9.1%  | 13.5% | 69.0% |
>
> As expected, combining DreamPRM with o4-mini achieves state-of-the-art on MathVista leaderboard:
>
> | Method           | Accuracy (%)|
> |------------------|-------------|
> | o4-mini (pass@1) | 80.6        |
> | Self-consistency | 81.7 (+1.1) |
> | DreamPRM         | 85.2 (+4.6) |
>
> **2. On pass@8 evaluation:**
> To further clarify the effect of test-time scaling, we report pass@8 results:
>
> | Benchmark   | pass@1 (%) | pass@8 (%) | Absolute Gain (%)|
> |-------------|---------|---------|----------------|
> | WeMath      | 51.7    | 78.4    | +26.7          |
> | MMStar      | 58.9    | 77.1    | +18.4          |
> | MathVista   | 65.4    | 77.4    | +12.0          |
> | MathVision  | 20.4    | 47.5    | +27.1          |
>
> However, pass@8 may overestimate model ability on benchmarks with multiple-choice questions (e.g., WeMath, MMStar, MathVista, MathVision). With few answer options, sampling diverse outputs increases the chance of guessing correctly — even if some responses follow flawed reasoning chains. Thus, pass@8 may not fully reflect reasoning capability.
>
> ---
> ***Response: Clarity of Figures and Analysis***
>
> **1. On radar chart interpretability:**
> In addition to the radar chart, we provide detailed quantitative results in Table 1 and Appendix Table 4. To improve clarity, we will modify the radar chart design in the revised version to better convey key differences.
>
> **2. Why certain datasets respond differently to ablations and scaling:**
> We analyze several representative cases from Figure 4.c and 5:
>
> - **MMVet w/o BLO and w/o AFL shows minimal drop (-0.2 and -1.4 respectively, shown in Figure 4.c):**
>   Both BLO and AFL relate to upper-level optimization. The limited drop is likely due to the low similarity between MMVet and the meta-learning dataset (MMMU). MMVet focuses on common-sense visual reasoning, which is less aligned with the content of MMMU. We quantify this using the angular difference $\Delta \theta$ (see metrics for inter-dataset similarity [2]) between the datasets:
>
>   $$
>   \Delta \theta = \frac{2}{\pi} \min \left[ \arccos(a_1 \cdot a_1'), \arccos(a_1 \cdot (-a_1')) \right]
>   $$
>
>   where $(a_1, a_1')$ is the first component vector of the datasets.
>
>   The lower the value of $\Delta \theta$, the more similar the datasets. The results are:
>
>   | Dataset Pair         | Angular Difference ($\Delta \theta$) | Interpretation   |
>   |----------------------|----------------------------------------|------------------|
>   | MMMU vs. MathVista   | 0.497                                  | More similar     |
>   | MMMU vs. MMVet       | 0.628                                  | Less similar     |
>
>   This confirms MMVet is less aligned with the meta-learning distribution, resulting in smaller gain from upper-level optimization.
>
> - **MMStar w/o ST drops the least (-0.7, shown in Figure 4.c):**
>   MMStar is constructed to assess tasks where visual information is essential. While ST explicitly encourages models to use vision, we find that models already rely on visual cues when necessary. As a result, removing ST leads to minimal degradation in performance (see MMStar benchmark [3]).
>
> - **MMVet has the largest and most unstable scaling variance (Figure 5):**
>   MMVet has only 218 samples, compared to ≥1000 for other datasets. Its smaller size results in higher variance and less stable performance when applying scaling strategies.
>
> ---
>
> ***Response: Inclusion of More Relevant Baselines***
>
> To address this, we include a direct comparison with VisualPRM [4] —a state-of-the-art multimodal PRM and the first (to our knowledge) to release a large-scale multimodal PRM dataset (VisualPRM400K, which is much larger than DreamPRM's 15K dataset).
>
> Below is a performance comparison between DreamPRM (ours) and VisualPRM:
>
> | Benchmark   | VisualPRM  (%) | DreamPRM  (%) |
> |-------------|------------|-----------------|
> | MathVision  | 21.3       | **22.1**        |
> | MMStar      | 61.4       | **62.3**        |
> | MMVet       | 61.2       | **61.4**        |
> | MathVista   | 68.5       | **68.9**        |
> | WeMath      | **59.1**   | 57.4            |
>
> As shown above, DreamPRM outperforms VisualPRM on 4 benchmarks. This illustrates the effectiveness of our domain-reweighted bi-level optimization framework. Unlike VisualPRM, which trains on all domains equally, DreamPRM adaptively emphasizes higher-quality domains during training. This design helps reduce quality imbalance and distribution shift, leading to better performance with fewer training samples.
>
> ---
> **References**
> [1] Wang et al., *Self-Consistency Improves Chain of Thought Reasoning in Language Models*, arXiv:2203.11171
> [2] Rajabinasab et al., *Metrics for Inter-Dataset Similarity with Example Applications in Synthetic Data and Feature Selection Evaluation*, SDM 2025
> [3] Chen et al., *Are We on the Right Way for Evaluating Large Vision-Language Models?*, arXiv:2403.20330
> [4] Wang et al.,*VisualPRM: An Effective Process Reward Model for Multimodal Reasoning*. arXiv:2503.10291.

---

> > ### Comment · Reviewer_Cvdg · 2025-08-04
> >
> > Thank you very much for your response and the additional information provided.
> >
> > First, I would like to point out that the paper proposes a PRM, so I believe the most important baselines should be the policy model and other PRMs. From the experimental results, we can see that o4-mini already ranks first (80.6%), so emphasizing that DreamPRM achieves first place in MathVisa is not particularly meaningful (the 2nd, 3rd, 4th, and 5th place methods use completely different base model capabilities).
> >
> > Second, I still cannot understand why self-consistency performs poorly. In mathematical reasoning, self-consistency can easily improve accuracy by 5-20% [1,2]. Therefore, it is quite surprising that in the experimental results, o4-mini + self-consistency only improves by 1.1\%.
> >
> > I have updated my rating based on all information
> >
> >
> > [1] DeepSeek-R1: Incentivizing Reasoning Capability in LLMs via Reinforcement Learning
> > [2] The Lessons of Developing Process Reward Models in Mathematical Reasoning

---

> ### Author Response · Authors · 2025-08-04
>
> Dear Reviewer Cvdg,
>
> Thank you again for your valuable feedback and questions.
> We wonder whether our responses have sufficiently addressed your comments?
> If more information is needed, we are very happy to provide it.
> We greatly appreciate your time and effort in this process.
>
> Best,
> Authors

---

> ### Author Response · Authors · 2025-08-04
> **Response to Reviewer Cvdg - Self-consistency Results (Part 1/2)**
>
> ## Response to Reviewer Cvdg: Part 1/2
>
> We appreciate your feedback. We would like to clarify several points regarding your comments on the self-consistency results and o4-mini performance.
>
> ---
> ***Response to "o4-mini + self-consistency only improves by 1.1%"***
>
> **1. Self-Consistency on MathVista Shows Small or Even Negative Gains in Prior Work:**
> The observed +1.1% improvement from self-consistency (o4-mini) is consistent with prior works on MathVista. In fact, multiple recent papers report similarly small gains for self-consistency@8:
>
> - **VisualPRM [1] (Table 6)**:
>   - InternVL-2.5-8B: **+1.4%**
>   - MiniCPM-v2.6: **–1.9%**
>
> - **AR-MCTS [2] (Table 1)**:
>   - GPT-4o: **+2.8%**
>   - LLaVA-OneVision-72B: **+1.8%**
>   - InternVL-2-8B: **+4.5%**
>   - Qwen2-VL-7B: **+2.4%**
>
> - **Our results**:
>   - o4-mini: **+1.1%**
>   - InternVL-2.5-8B-MPO: **+1.7%**
>
> These results show that self-consistency provides only **marginal improvements** in the MathVista benchmark across a wide range of models, and in some cases may even **reduce** performance.
>
> ---
> ***Response to "self-consistency can easily improve accuracy by 5-20% [3,4]."***
> **1. DeepSeek-R1 [3] Comparison is Different Due to Limited Benchmark Size and Larger Sampling Scale**
> We understand the reviewer’s expectation that self-consistency may lead to significant performance gains. However, we respectfully note that comparing our **self-consistency@8 on multimodal reasoning benchmarks** with DeepSeek's **self-consistency@64 on AIME 2024 benchmark** is not an apples-to-apples comparison. The two differ in several critical aspects:
>
> - **Task nature**: DeepSeek's benchmark AIME 2024 contains **only 30 questions**, so a 20% improvement corresponds to 6 questions. Given the small size of the benchmark, such results are naturally subject to higher variance.
> - **Sampling budget**: DeepSeek reports **cons@64**, which naturally has a higher upper bound than **cons@8**.
> - **Modality**: Our benchmarks are multimodal (e.g., MathVista), where the capabilities of models are different.
>
> We also recognized that the effectiveness of self-consistency varies depending on these factors. To better understand this variation, we conducted a detailed analysis of the **distribution of correct CoTs** in the above rebuttal's **Response: Analysis of Test-Time Scaling** part. We believe this analysis is highly relevant to understanding when and why self-consistency performs well.
>
> Moreover, this distribution analysis provides a **transparent and interpretable statistical view** of how our results are obtained. In fact, one can **approximate the expected self-consistency performance** by summing the proportion of samples with **4–7 correct CoTs** and those with **all 8 correct CoTs**.
>
>
> **2. Self-Consistency Gains in Qwen2.5-Math-PRM [4] Also Vary Substantially Across Tasks**
> We acknowledge your observation regarding the potential effectiveness of self-consistency. According to the Qwen2.5-Math-PRM paper (Table 8), the self-consistency gains (maj@8 vs. pass@1) are as follows:
>
> | Benchmark          | pass@1 | maj@8 | Gain |
> |--------------------|--------|--------|------|
> | GSM8K              | 91.2   | 93.7   | +2.5 |
> | MATH               | 74.0   | 80.3   | +6.3 |
> | Minerva Math       | 32.0   | 37.1   | +5.1 |
> | Gaokao 2023 En     | 64.7   | 69.9   | +5.2 |
> | Olympiad Bench     | 36.9   | 45.8   | +8.9 |
> | College Math       | 46.2   | 48.5   | +2.3 |
> | MMLU STEM          | 57.1   | 61.9   | +4.8 |
>
> All reported improvements fall within a **2%–9%** range. This highlights that the impact of self-consistency is **highly task-specific**. Our findings on multimodal benchmarks follow the same trend, suggesting that such variation is expected and depends on task structure and model behavior.
>
> **(o4-mini performance related question is addressed in the following comment due to character limit)**
>
> [1] Wang et al.,*VisualPRM: An Effective Process Reward Model for Multimodal Reasoning*. arXiv:2503.10291.
> [2] Dong et al. *Progressive multimodal reasoning via active retrieval.* arXiv:2412.14835.
> [3] Guo et al. *Deepseek-r1: Incentivizing reasoning capability in llms via reinforcement learning.* arXiv:2501.12948.
> [4] Zhang et al. *The lessons of developing process reward models in mathematical reasoning.* arXiv:2501.07301.

---

> ### Author Response · Authors · 2025-08-04
> **Response to Reviewer Cvdg - o4-mini Performance (Part 2/2)**
>
> ## Response to Reviewer Cvdg: Part 2/2
>
> We appreciate your feedback. We would like to clarify several points regarding your comments on the self-consistency results and o4-mini performance.
>
> ---
> ***Response to "the most important baselines should be the policy model and other PRMs"***
>
> We would like to clarify the motivation behind including **o4-mini** results in the Common Response. Our goal was **not** to change the base model for performance gain, but rather to **demonstrate the generalizability of our PRM** — showing that it can effectively enhance the performance of **strong, recent models** such as o4-mini. This strengthens our claim that DreamPRM is generalizable and broadly applicable.
>
> Also, we agree that comparisons with other PRM-based methods are important, so below, we provide a direct comparison between **DreamPRM** and 5 reward model baselines (including **VisualPRM [1]**) using o4-mini on the MathVista benchmark:
>
> | Method                         | Accuracy (cons@8) |
> |--------------------------------|-------------------|
> | o4-mini (pass@1)               | 80.6              |
> | o4-mini + ORM (8 CoTs)         | 82.8              |
> | o4-mini + Vanilla-PRM (8 CoTs) | 84.2              |
> | o4-mini + s1-PRM (8 CoTs)      | 83.5              |
> | o4-mini + CaR-PRM (8 CoTs)     | 83.9              |
> | o4-mini + VisualPRM (8 CoTs)   | 84.1              |
> | o4-mini + DreamPRM (8 CoTs)    | **85.2**          |
>
> As shown, DreamPRM outperforms other baselines by a clear margin even on top-tier models. This result supports the effectiveness and **generalization ability** of our method — not only on InternVL, but also on the latest, strongest multimodal large language models.
>
> In the main paper, we evaluate DreamPRM across **five multimodal reasoning benchmarks** using **InternVL** as the primary base model, and also provide results with **GPT-4.1** on one benchmark to show potential under stronger language models. During the rebuttal phase, we further strengthened the empirical evaluation by:  (1) adding **two new benchmarks** (also with InternVL),  (2) including **VisualPRM** as an additional strong baseline for comparison, and  (3) evaluating **o4-mini** on two benchmarks, which achieves **state-of-the-art results**. In these settings, DreamPRM consistently outperforms strong baselines, demonstrating its robustness across diverse tasks and backbones.
>
> [1] Wang et al.,*VisualPRM: An Effective Process Reward Model for Multimodal Reasoning*. arXiv:2503.10291.

---

### Note · Authors · 2025-08-11

Dear AC and reviewers,

Thank you for reviewing our paper. We would like to highlight two key points in our final remarks.

1. **On self-consistency gains**
Empirically, multiple recent works on MathVista report small or even negative improvements for self-consistency@8 (typically –2% to +4%). The high gain reported in DeepSeek’s self-consistency@64 comes from a 30-question, text-only benchmark, which does not imply that our large-scale (mostly ≥1000 questions) multimodal benchmarks with self-consistency@8 should achieve similar gains. Our +1.1% result is fully consistent with the empirical trend, and we are confident that the self-consistency baseline in our evaluation was implemented correctly.

2. **On PRM baselines**
We have included as many PRM-based baselines as were reasonably available: four PRM methods and one ORM method under the o4-mini setting, and four PRM/ORM baselines under the InternVL setting, all evaluated across five benchmarks. DreamPRM outperforms every baseline in all settings. In our rebuttal, we also added **VisualPRM** — the widely recognized multimodal PRM — as requested, and our method continued to show clear advantages.

Finally, we want to note that to the best of our knowledge, our work is the **first to introduce domain reweighting and bi-level optimization into the reasoning domain**, opening a new methodological direction for improving multimodal reasoning.
We also conducted comprehensive evaluation, covering **8 strong baselines** (policy model, self-consistency, self-correction, ORM, Vanilla PRM, s1-PRM, CaR-PRM, VisualPRM), **7 multimodal benchmarks** (MathVista, WeMath, MathVision, MMVet, MMStar, SciVerse, OlympiadBench), and **3 latest policy models** (InternVL-2.5-8B-MPO, GPT-4.1, o4-mini), along with ablation studies, generalization tests, scaling analyses, training dynamics, and case studies.

We appreciate the AC’s and reviewers’ time and consideration of these points in the final decision process.

Best regards,
Authors

---

### Decision · Program_Chairs · 2025-09-17

**Decision:**

Accept (poster)

**Comment:**

This paper proposes DreamPRM, a domain-reweighted Process Reward Model for multimodal reasoning. DreamPRM formulates PRM training as a bi-level optimization problem, in which a lower level fine-tunes the PRM with domain weights while an upper level adjusts those weights on a meta-dataset using an aggregation loss aligned with inference-time selection. The reviewers found the work well-motivated, paper clearly written, and method of introducing domain reweighting and bi-level optimization to PRMs as novel. They were also convinced about the effectiveness of the method, which was supported by extensive experiments across diverse multimodal benchmarks and baselines. Reviewers initially raised concerns about possible data leakage into MathVista, small self-consistency gains, and the need for stronger baselines. However, during the rebuttal, they were largely addressed through filtering, additional analyses, and expanded experiments, and all reviewers leaned toward acceptance. Yet, the work has remaining limitations include reliance on ground-truth answers for Monte Carlo step annotation, the computational cost and potential suboptimality of approximate bi-level optimization, and sensitivity to meta-set design, which made the paper fall into a borderline case. Overall, despite these limitations, the method is technically solid, introduces a novel and useful approach to improving multimodal PRMs, and thus I recommend acceptance.